# SayPlan: Grounding Large Language Models using 3D Scene Graphs for Scalable Robot Task Planning

**Krishan Rana**[†1], **Jesse Haviland**[*1,2], **Sourav Garg**[*3], **Jad Abou-Chakra**[*1],
**Ian Reid**[3], **Niko Sünderhauf**[1]

[1]QUT Centre for Robotics, Queensland University of Technology
[2]CSIRO Data61 Robotics and Autonomous Systems Group
[3]University of Adelaide
[*]Equal Contribution
[†]`ranak@qut.edu.au`

**Abstract:**

Large language models (LLMs) have demonstrated impressive results in developing generalist planning agents for diverse tasks. However, grounding these plans in expansive, multi-floor, and multi-room environments presents a significant challenge for robotics. We introduce SayPlan, a scalable approach to LLM-based, large-scale task planning for robotics using 3D scene graph (3DSG) representations. To ensure the scalability of our approach, we: (1) exploit the hierarchical nature of 3DSGs to allow LLMs to conduct a *semantic search* for task-relevant subgraphs from a smaller, collapsed representation of the full graph; (2) reduce the planning horizon for the LLM by integrating a classical path planner and (3) introduce an *iterative replanning* pipeline that refines the initial plan using feedback from a scene graph simulator, correcting infeasible actions and avoiding planning failures. We evaluate our approach on two large-scale environments spanning up to 3 floors and 36 rooms with 140 assets and objects and show that our approach is capable of grounding large-scale, long-horizon task plans from abstract, and natural language instruction for a mobile manipulator robot to execute. We provide real robot video demonstrations on our project page sayplan.github.io.

## 1 Introduction

*"Make me a coffee and place it on my desk"* – The successful execution of such a seemingly straightforward command remains a daunting task for today's robots. The associated challenges permeate every aspect of robotics, encompassing navigation, perception, manipulation as well as high-level task planning. Recent advances in Large Language Models (LLMs) [1, 2, 3] have led to significant progress in incorporating common sense knowledge for robotics [4, 5, 6]. This enables robots to plan complex strategies for a diverse range of tasks that require a substantial amount of background knowledge and semantic comprehension.

For LLMs to be effective planners in robotics, they must be grounded in reality, that is, they must adhere to the constraints presented by the physical environment in which the robot operates, including the available affordances, relevant predicates, and the impact of actions on the current state. Furthermore, in expansive environments, the robot must additionally understand where it is, locate items of interest, as well comprehend the topological arrangement of the environment in order to plan across the necessary regions. To address this, recent works have explored the utilization of vision-based value functions [4], object detectors [7, 8], or Planning Domain Definition Language (PDDL) descriptions of a scene [9, 10] to ground the output of the LLM-based planner. However, these efforts are primarily confined to small-scale environments, typically single rooms with pre-encoded information on all the existing assets and objects present. The challenge lies in scaling these models. As the environment's complexity and dimensions expand, and as more rooms and entities enter the

7th Conference on Robot Learning (CoRL 2023), Atlanta, USA.

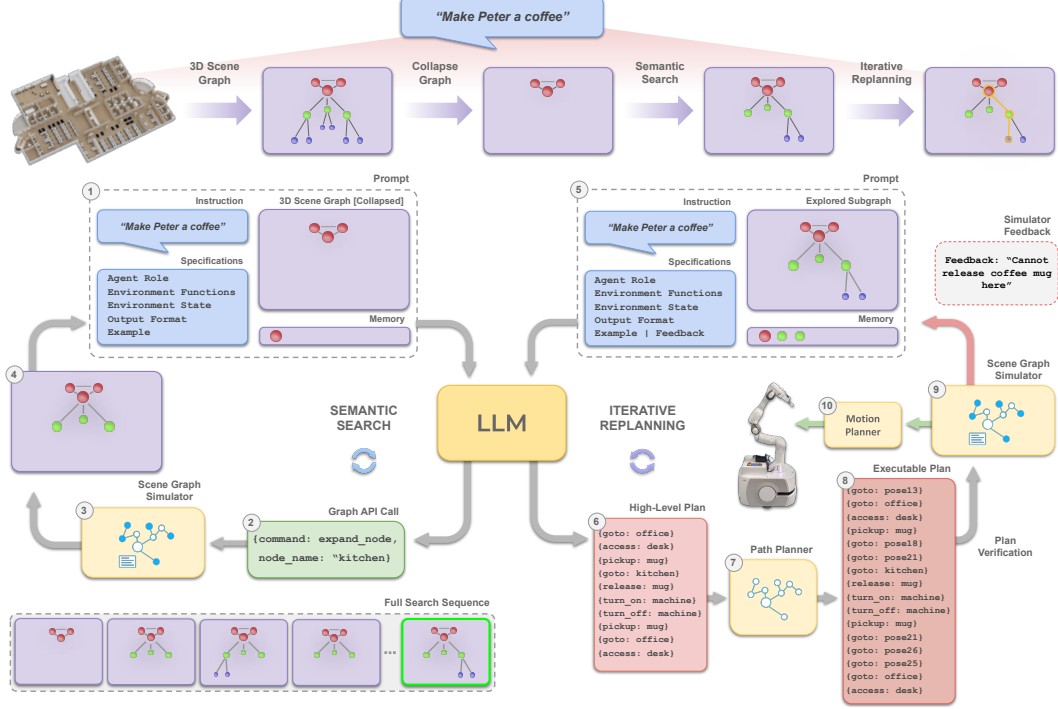

Figure 1: **SayPlan Overview (top).** SayPlan operates across two stages to ensure scalability: (left) Given a collapsed 3D scene graph and a task instruction, *semantic search* is conducted by the LLM to identify a suitable subgraph that contains the required items to solve the task; (right) The explored subgraph is then used by the LLM to generate a high-level task plan, where a classical path planner completes the navigational component of the plan; finally, the plan goes through an *iterative replanning* process with feedback from a scene graph simulator until an executable plan is identified. Numbers on the top-left corners represent the flow of operations.

scene, pre-encoding all the necessary information within the LLM's context becomes increasingly infeasible.

To this end, we present a scalable approach to ground LLM-based task planners across environments spanning multiple rooms and floors. We achieve this by exploiting the growing body of 3D scene graph (3DSG) research [11, 12, 13, 14, 15, 16]. 3DSGs capture a rich topological and hierarchically-organised semantic graph representation of an environment with the versatility to encode the necessary information required for task planning including object state, predicates, affordances and attributes using natural language – suitable for parsing by an LLM. We can leverage a JSON representation of this graph as input to a pre-trained LLM, however, to ensure the *scalability* of the plans to expansive scenes, we present three key innovations.

Firstly, we present a mechanism that enables the LLM to conduct a *semantic search* for a task-relevant subgraph $\mathcal{G}'$ by manipulating the nodes of a 'collapsed' 3DSG, which exposes only the top level of the full graph $\mathcal{G}$, via `expand` and `contract` API function calls – thus making it feasible to plan over increasingly large-scale environments. In doing so, the LLM maintains focus on a relatively small, informative subgraph, $\mathcal{G}'$ during planning, without exceeding its token limit. Secondly, as the horizon of the task plans across such environments tends to grow with the complexity and range of the given task instructions, there is an increasing tendency for the LLM to hallucinate or produce infeasible action sequences [17, 18, 7]. We counter this by firstly relaxing the need for the LLM to generate the navigational component of the plan, and instead leverage an existing optimal path planner such as Dijkstra [19] to connect high-level nodes generated by the LLM. Finally, to ensure the feasibility of the proposed plan, we introduce an *iterative replanning* pipeline that verifies and refines the initial plan using feedback from a *scene graph simulator* in order to correct for any unexecutable actions, e.g., missing to open the fridge before putting something into it – thus avoiding planning failures due to inconsistencies, hallucinations, or violations of the physical constraints and predicates imposed by the environment.

Our approach SayPlan ensures feasible and grounded plan generation for a mobile manipulator robot operating in large-scale environments spanning multiple floors and rooms. We evaluate our framework across a range of 90 tasks organised into four levels of difficulty. These include semantic search tasks such as (*"Find me something non-vegetarian."*) to interactive, long-horizon tasks with ambiguous multi-room objectives that require a significant level of common-sense reasoning (*"Let's play a prank on Niko"*). These tasks are assessed in two expansive environments, including a large office floor spanning 37 rooms and 150 interactable assets and objects, and a three-storey house with 28 rooms and 112 objects. Our experiments validate SayPlan's ability to scale task planning to large-scale environments while conserving a low token footprint. By introducing a semantic search pipeline, we can reduce full large-scale scene representations by up to 82.1% for LLM parsing and our iterative replanning pipeline allows for near-perfect executability rates, suitable for execution on a real mobile manipulator robot.[1]

## 2 Related Work

**Task planning in robotics** aims to generate a sequence of high-level actions to achieve a goal within an environment. Conventional methods employ domain-specific languages such as PDDL [20, 21, 22] and ASP [23] together with semantic parsing [24, 25], search techniques [26, 27] and complex heuristics [28] to arrive at a solution. These methods, however, lack both the scalability to large environments as well as the task generality required when operating in the real world. Hierarchical and reinforcement learning-based alternatives [29, 30], [31] face challenges with data demands and scalability. Our work leverages the in-context learning capabilities of LLMs to generate task plans across 3D scene graphs. Tasks, in this case, can be naturally expressed using language, with the internet scale training of LLMs providing the desired knowledge for task generality, while 3D scene graphs provide the grounding necessary for large-scale environment operation. This allows for a general and scalable framework when compared to traditional non-LLM-based alternatives.

**Task planning with LLMs**, that is, translating natural language prompts into task plans for robotics, is an emergent trend in the field. Earlier studies have effectively leveraged pre-trained LLMs' in-context learning abilities to generate actionable plans for embodied agents [4, 10, 9, 8, 32, 7, 33]. A key challenge for robotics is grounding these plans within the operational environment of the robot. Prior works have explored the use of object detectors [8, 7], PDDL environment representations [10, 9, 34] or value functions [4] to achieve this grounding, however, they are predominantly constrained to single-room environments, and scale poorly with the number of objects in a scene which limits their ability to plan over multi-room or multi-floor environments. In this work, we explore the use of 3D scene graphs and the ability of LLMs to generate plans over large-scale scenes by exploiting the inherent hierarchical and semantic nature of these representations.

**Integrating external knowledge in LLMs** has been a growing line of research combining language models with external tools to improve the reliability of their outputs. In such cases, external modules are used to provide feedback or extra information to the LLM to guide its output generation. This is achieved either through API calls to external tools [35, 36] or as textual feedback from the operating environment [37, 8]. More closely related to our work, CLAIRIFY [38] iteratively leverage compiler error feedback to re-prompt an LLM to generate syntactically valid code. Building on these ideas, we propose an iterative plan verification process with feedback from a scene graph-based simulator to ensure all generated plans adhere to the constraints and predicates captured by the pre-constructed scene graph. This ensures the direct executability of the plan on a mobile manipulator robot, operating in the corresponding real-world environment.

## 3 SayPlan

### 3.1 Problem Formulation

We aim to address the challenge of long-range task planning for an autonomous agent, such as a mobile manipulator robot, in a large-scale environment based on natural language instructions. This requires the robot to comprehend abstract and ambiguous instructions, understand the scene and generate task plans involving both navigation and manipulation of a mobile robot within an

---

[1]sayplan.github.io

**Algorithm 1:** SayPlan

**Given:** scene graph simulator $\psi$, classical path planner $\phi$, large language model $LLM$
**Inputs:** prompt $\mathcal{P}$, scene graph $\mathcal{G}$, instruction $\mathcal{I}$

1: $\mathcal{G}' \leftarrow \texttt{collapse}_\psi(\mathcal{G})$        ▷ collapse scene graph
    **Stage 1:** Semantic Search      ▷ search scene graph for all relevant items
2:  **while** command != *"terminate"* **do**
3:    command, node_name $\leftarrow LLM(\mathcal{P}, \mathcal{G}', \mathcal{I})$
4:    **if** command == *"expand"* **then**
5:      $\mathcal{G}' \leftarrow \texttt{expand}_\psi$(node_name)      ▷ expand node to reveal objects and assets
6:    **else if** command == *"contract"* **then**
7:      $\mathcal{G}' \leftarrow \texttt{contract}_\psi$(node_name)      ▷ contract node if nothing relevant found
    **Stage 2:** Causal Planning      ▷ generate a feasible plan
8:  feedback = " "
9:  **while** feedback != *"success"* **do**
10:   plan $\leftarrow LLM(\mathcal{P}, \mathcal{G}', I, \text{feedback})$      ▷ high level plan
11:   full_plan $\leftarrow \phi(\text{plan}, \mathcal{G}')$      ▷ compute optimal navigational path between nodes
12:   feedback $\leftarrow \texttt{verify\_plan}_\psi$(full_plan)      ▷ forward simulate the full plan
13: **return** full_plan      ▷ executable plan

environment. Existing approaches lack the ability to reason over scenes spanning multiple floors and rooms. Our focus is on integrating large-scale scenes into planning agents based on Language Models (LLMs) and solving the scalability challenge. We aim to tackle two key problems: 1) representing large-scale scenes within LLM token limitations, and 2) mitigating LLM hallucinations and erroneous outputs when generating long-horizon plans in large-scale environments.

### 3.2 Preliminaries

Here, we describe the 3D scene graph representation of an environment and the scene graph simulator API which we leverage throughout our approach.

**Scene Representation:** 3D Scene Graphs (3DSG) [11, 12, 14] have recently emerged as an actionable world representation for robots [13, 15, 16, 39, 40, 41], which hierarchically abstract the environment at multiple levels through spatial semantics and object relationships while capturing relevant states, affordances and predicates of the entities present in the environment. Formally, a 3DSG is a hierarchical multigraph $\mathcal{G} = (V, E)$ in which the set of vertices $V$ comprises $V_1 \cup V_2 \cup \ldots \cup V_K$, with

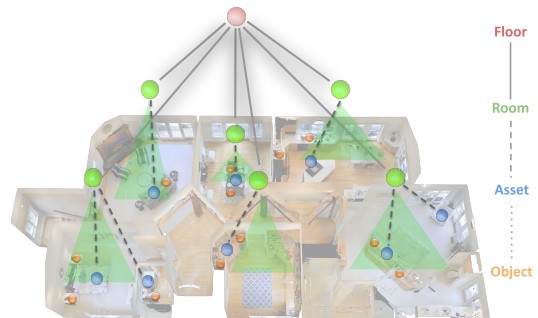

Figure 2: **Hierarchical Structure of a 3D Scene Graph.** This graph consists of 4 levels. Notes that the room nodes are connected to one another via sequences of pose nodes which capture the topological arrangement of a scene.

each $V_k$ signifying the set of vertices at a particular level of the hierarchy $k$. Edges stemming from a vertex $v \in V_k$ may only terminate in $V_{k-1} \cup V_k \cup V_{k+1}$, i.e. edges connect nodes within the same level, or one level higher or lower.

We assume a pre-constructed 3DSG representation of a large-scale environment generated using existing techniques [15, 13, 11]. The entire 3DSG can be represented as a NetworkX `Graph` object [42] and text-serialised into a JSON data format that can be parsed directly by a pre-trained LLM. An example of a single asset node from the 3DSG is represented as: {`name: coffee_machine, type: asset, location: kitchen, affordances: [turn_on, turn_off, release], state: off, attributes: [red, automatic], position: [2.34, 0.45, 2.23]`} with edges between nodes captured as {`kitchen↔coffee_machine`}. The 3DSG is organized in a hierarchical manner with four primary levels: floors, rooms, assets, and objects as shown in Figure 2. The top level contains floors, each of which branches out to several rooms. These rooms are interconnected through pose nodes to represent the environment's topological structure. Within each room, we find assets (immovable entities) and objects (movable entities). Both asset and object nodes encode particulars including state, affordances, additional attributes such as colour or weight, and 3D pose. The graph also incorporates a dynamic agent

node, denoting a robot's location within the scene. Note that this hierarchy is scalable and node levels can be adapted to capture even larger environments e.g. campuses and buildings

**Scene Graph Simulator** $\psi$ refers to a set of API calls for manipulating and operating over JSON formatted 3DSGs, using the following functions: 1) `collapse`$(\mathcal{G})$ : Given a full 3DSG, this function returns an updated scene graph that exposes only the highest level within the 3DSG hierarchy e.g. floor nodes. 2) `expand(node_name)` : Returns an updated 3DSG that reveals all the nodes connected to `node_name` in the level below. 3) `contract(node_name)` : Returns an updated 3DSG that hides all the nodes connected to `node_name` in the level below. 4) `verify_plan(plan)` : Forward simulates the generated plan at the abstract graph level captured by the 3DSG to check if each action adheres to the environment's predicates, states and affordances. Returns textual feedback e.g. *"cannot pick up banana"* if the fridge containing the banana is closed.

### 3.3 Approach

We present a scalable framework for grounding the generalist task planning capabilities of pre-trained LLMs in large-scale environments spanning multiple floors and rooms using 3DSG representations. Given a 3DSG $\mathcal{G}$ and a task instruction $\mathcal{I}$ defined in natural language, we can view our framework SayPlan as a high-level task planner $\pi(\boldsymbol{a}|\mathcal{I}, \mathcal{G})$, capable of generating long-horizon plans $\boldsymbol{a}$ grounded in the environment within which a mobile manipulator robot operates. This plan is then fed to a low-level visually grounded motion planner for real-world execution. To ensure the scalability of SayPlan, two stages are introduced: *Semantic Search* and *Iterative Replanning* which we detail below. An overview of the SayPlan pipeline is illustrated in Figure 1 with the corresponding pseudo-code given in Algorithm 1.

**Semantic Search:** When planning over 3DSGs using LLMs we take note of two key observations: **1)** A 3DSG of a large-scale environment can grow infinitely with the number of rooms, assets and objects it contains, making it impractical to pass as input to an LLM due to token limits and **2)** only a subset of the full 3DSG $\mathcal{G}$ is required to solve any given task e.g. we don't need to know about the toothpaste in the bathroom when making a cup of coffee. To this end, the Semantic Search stage seeks to identify this smaller, task-specific subgraph $\mathcal{G}'$ from the full 3DSG which only contains the entities in the environment required to solve the given task instruction. To identify $\mathcal{G}'$ from a full 3DSG, we exploit the semantic hierarchy of these representations and the reasoning capabilities of LLMs. We firstly `collapse` $\mathcal{G}$ to expose only its top level e.g. the floor nodes, reducing the 3DSG initial token representation by $\approx 80\%$. The LLM manipulates this collapsed graph via `expand` and `contract` API calls in order to identify the desired subgraph for the task based on the given instruction $\mathcal{I}$. This is achieved using in-context learning over a set of input-out examples (see Appendix J), and utilising chain-of-thought prompting to guide the LLM in identifying which nodes to manipulate. The chosen API call and node are executed within the scene graph simulator, and the updated 3DSG is passed back to the LLM for further exploration. If an expanded node is found to contain irrelevant entities for the task, the LLM contracts it to manage token limitations and maintain a task-specific subgraph (see Figure 3). To avoid expanding already-contracted nodes, we maintain a list of previously expanded nodes, passed as an additional **Memory** input to the LLM, facilitating a Markovian decision-making process and allowing SayPlan to scale to extensive search sequences without the overhead of maintaining the full interaction history [5]. The LLM autonomously proceeds to the planning phase once all necessary assets and objects are identified in the current subgraph $\mathcal{G}'$. An example of the LLM-scene graph interaction during Semantic Search is provided in Appendix K.

**Iterative Replanning:** Given the identified subgraph $\mathcal{G}'$ and the same task instruction $\mathcal{I}$ from above, the LLM enters the planning stage of the pipeline. Here the LLM is tasked with generating a sequence of node-level navigational (`goto(pose2)`) and manipulation (`pickup(coffee_mug)`) actions that satisfy the given task instruction. LLMs, however, are not perfect planning agents and tend to hallucinate or produce erroneous outputs [43, 9]. This is further exacerbated when planning over large-scale environments or long-horizon tasks. We facilitate the generation of task plans by the LLM via two mechanisms. First, we shorten the LLM's planning horizon by delegating pose-level path planning to an optimal path planner, such as Dijkstra. For example, a typical plan output such as `[goto(meeting_room), goto(pose13), goto(pose14), goto(pose8), ..., goto(kitchen), access(fridge), open(fridge)]` is simplified to `[goto(meeting_room), goto(kitchen), access(fridge), open(fridge)]`. The path

planner handles finding the optimal route between high-level locations, allowing the LLM to focus on essential manipulation components of the task. Secondly, we build on the self-reflection capabilities of LLMs [17] to iteratively correct their generated plans using textual, task-agnostic feedback from a `scene graph simulator` which evaluates if the generated plan complies with the scene graph's predicates, state, and affordances. For instance, a `pick(banana)` action might fail if the robot is already holding something, if it is not in the correct location or if the fridge was not opened beforehand. Such failures are transformed into textual feedback (e.g., *"cannot pick banana"*), appended to the LLM's input, and used to generate an updated, executable plan. This iterative process, involving planning, validation, and feedback integration, continues until a feasible plan is obtained. The validated plan is then passed to a low-level motion planner for robotic execution. An example of the LLM-scene graph interaction during iterative replanning is provided in Appendix L. Specific implementation details are provided in Appendix A.

## 4    Experimental Setup

We design our experiments to evaluate the 3D scene graph reasoning capabilities of LLMs with a particular focus on high-level task planning pertaining to a mobile manipulator robot. The plans adhere to a particular embodiment consisting of a 7-degree-of-freedom robot arm with a two-fingered gripper attached to a mobile base. We use two large-scale environments, shown in Figure 4, which exhibit multiple rooms and multiple floors which the LLM agent has to plan across. To better ablate and showcase the capabilities of SayPlan, we decouple its semantic search ability from the overall causal planning capabilities using the following two evaluation settings as shown in Appendix C:

**Semantic Search:**    Here, we focus on queries which test the semantic search capabilities of an LLM provided with a collapsed 3D scene graph. This requires the LLM to reason over the room and floor node names and their corresponding attributes in order to aid its search for the relevant assets and objects required to solve the given task instruction. We evaluate against a human baseline to understand how the semantic search capabilities of an LLM compare to a human's thought process. Furthermore, to gain a better understanding of the impact different LLM models have on this graph-based reasoning, we additionally compare against a variant of SayPlan using `GPT-3.5`.

**Causal Planning:**    In this experiment, we evaluate the ability of SayPlan to generate feasible plans to solve a given natural language instruction. The evaluation metrics are divided into two components: *1) Correctness*, which primarily validates the overall goal of the plan and its alignment to what a human would do to solve the task and *2) Executability*, which evaluates the alignment of the plan to the constraints of the scene graph environment and its ability to be executed by a mobile manipulator robot. We note here that for a plan to be executable, it does not necessarily have to be correct and vice versa. We evaluate SayPlan against two baseline methods that integrate an LLM for task planning:

**LLM-As-Planner**, which generates a full plan sequence in an open-loop manner; the plan includes the full sequence of both navigation and manipulation actions that the robot must execute to complete a task, and **LLM+P**, an ablated variant of SayPlan, which only incorporates the path planner to allow for shorter horizon plan sequences, without any iterative replanning.

## 5    Results

### 5.1    Semantic Search

We summarise the results for the semantic search evaluation in Table 1. SayPlan (`GPT-3.5`) consistently failed to reason over the input graph representation, hallucinating nodes to explore or stagnating at exploring the same node multiple times. SayPlan (`GPT-4`) in contrast achieved 86.7% and 73.3% success in identifying the desired subgraph across both the simple and complex search tasks respectively, demonstrating significantly better graph-based reasoning than `GPT-3.5`.

|  | Office | | | Home | | |
|---|---|---|---|---|---|---|
| Subtask | Human | SayPlan (`GPT-3.5`) | SayPlan (`GPT-4`) | Human | SayPlan (`GPT-3.5`) | SayPlan (`GPT-4`) |
| **Simple Search** | 100% | 6.6% | 86.7% | 100% | 0.0% | 86.7% |
| **Complex Search** | 100% | 0.0% | 73.3% | 100% | 0.0% | 73.3% |

Table 1: **Evaluating the semantic search capabilities of `GPT-4`.** The table shows the semantic search success rate in finding a suitable subgraph for planning.

| | Simple | | Long Horizon | | Types of Errors | | | | |
|---|---|---|---|---|---|---|---|---|---|
| | **Corr** | **Exec** | **Corr** | **Exec** | **Missing Action** | **Missing Pose** | **Wrong Action** | **Incomplete Search** | **Hallucinated Nodes** |
| **LLM+P** | 93.3% | 13.3% | 33.3% | 0.0% | 26.7% | 0.0% | 10.0% | 3.33% | 10.0% |
| **LLM-As-Planner** | 93.3% | 80.0% | 66.7% | 13.3% | 20.0% | 60.0% | 0.17% | 0.03% | 10.0% |
| **SayPlan** | 93.3% | 100.0% | 73.3% | 86.6% | 0.0% | 0.0% | 0.0% | 0.0% | 6.67% |

Table 3: **Causal Planning Results.** *Left:* **Corr**ectness and **Exec**utability on Simple and Long Horizon planning tasks and *Right:* Types of execution errors encountered when planning using LLMs. Note that SayPlan corrects the majority of the errors faced by LLM-based planners.

While as expected the human baseline achieved 100% on all sets of instructions, we are more interested in the qualitative assessment of the common-sense reasoning used during semantic search. More specifically we would like to identify the similarity in the semantic search heuristics utilised by humans and that used by the underlying LLM based on the given task instruction.

We present the full sequence of explored nodes for both SayPlan (GPT-4) and the human baseline in Appendix F. As shown in the tables, SayPlan (GPT-4) demonstrates remarkably similar performance to a human's semantic and common sense reasoning for most tasks, exploring a similar sequence of nodes given a particular instruction. For example, when asked to *"find a ripe banana"*, the LLM first explores the kitchen followed by the next most likely location, the cafeteria. In the case where no semantics are present in the instruction such as *"find me object K31X"*, we note that the LLM agent is capable of conducting a breadth-first-like search across all the unexplored nodes. This highlights the importance of meaningful node names and attributes that capture the relevant environment semantics that the LLM can leverage to relate the query instruction for efficient search.

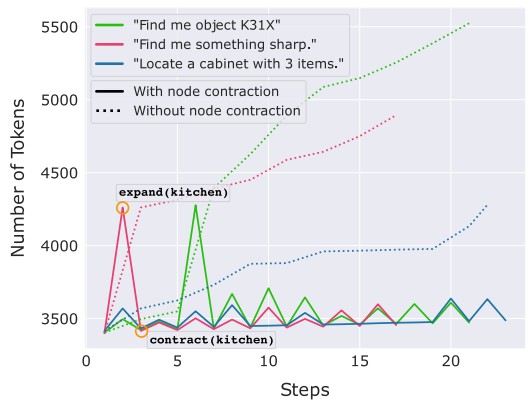

Figure 3: **Scene Graph Token Progression During Semantic Search.** This graph illustrates the scalability of our approach to large-scale 3D scene graphs. Note the importance of node contraction in maintaining a near constant token representation of the 3DSG input.

| | **Full Graph** (Token Count) | **Collapsed Graph** (Token Count) | **Compression Ratio** |
|---|---|---|---|
| **Office** | 6731 | 878 | 86.9% |
| **Home** | 6598 | 1817 | 72.5% |

Table 2: **3D Scene Graph Token Count** Number of tokens required for the full graph vs. collapsed graph.

An odd failure case in the simple search instructions involved negation, where the agent consistently failed when presented with questions such as *"Find me an office that does not have a cabinet"* or *"Find me a bathroom with no toilet"*. Other failure cases noted across the complex search instructions included the LLM's failure to conduct simple distance-based and count-based reasoning over graph nodes. While trivial to a human, this does require the LLM agent to reason over multiple nodes simultaneously, where it tends to hallucinate or miscount connected nodes.

**Scalability Analysis:** We additionally analyse the scalability of SayPlan during semantic search. Table 2 illustrates the impact of exploiting the hierarchical nature of 3D scene graphs and allowing the LLM to explore the graph from a collapsed initial state. This allows for a reduction of 82.1% in the initial input tokens required to represent the Office environment and a 60.4% reduction for the Home environment. In Figure 3, we illustrate how endowing the LLM with the ability to contract explored nodes which it deems unsuitable for solving the task allows it to maintain near-constant input memory from a token perspective across the entire semantic search process. Note that the initial number of tokens already present represents the input prompt tokens as given in Appendix J. Further ablation studies on the scalability of SayPlan to even larger 3DSGs are provided in Appendix H.

### 5.2 Causal Planning

The results for causal planning across simple and long-horizon instructions are summarised in Table 3 (left). We compared SayPlan's performance against two baselines: LLM-As-Planner and LLM+P. All three methods displayed consistent correctness in simple planning tasks at 93%, given that this metric is more a function of the underlying LLMs reasoning capabilities. However, it is interesting to note that in the long-horizon tasks, both the path planner and iterative replanning play an important role in improving this correctness metric by reducing the planning horizon and allowing the LLM to reflect on its previous output.

The results illustrate that the key to ensuring the task plan's executability was iterative replanning. Both LLM-As-Planner and LLM+P exhibited poor executability, whereas SayPlan achieved near-perfect executability as a result of iterative replanning, which ensured that the generated plans were grounded to adhere to the constraints and predicated imposed by the environment. Detailed task plans and errors encountered are provided in Appendix G. We summarise these errors in Table 3 (right) which shows that plans generated with LLM+P and LLM-As-Planner entailed various types of errors limiting their executability. LLM+P mitigated navigational path planning errors as a result of the classical path planner however still suffered from errors pertaining to the manipulation of the environment - missing actions or incorrect actions which violate environment predicates. SayPlan mitigated these errors via iterative replanning, however in 6.67% of tasks, it failed to correct for some hallucinated nodes. While we believe these errors could be eventually corrected via iterative replanning, we limited the number of replanning steps to 5 throughout all experiments. We provide an illustration of the real-world execution of a generated plan using SayPlan on a mobile manipulator robot coupled with a vision-guided motion controller [44, 45] in Appendix I.

## 6 Limitations

SayPlan is notably constrained by the limitations inherent in current large language models (LLMs), including biases and inaccuracies, affecting the validity of its generated plans. More specifically, SayPlan is limited by the graph-based reasoning capabilities of the underlying LLM which fails at simple distance-based reasoning, node count-based reasoning and node negation. Future work could explore fine-tuning these models for these specific tasks or alternatively incorporate existing and more complex graph reasoning tools [46] to facilitate decision-making. Secondly, SayPlan's current framework is constrained by the need for a pre-built 3D scene graph and assumes that objects remain static post-map generation, significantly restricting its adaptability to dynamic real-world environments. Future work could explore how online scene graph SLAM systems [15] could be integrated within the SayPlan framework to account for this. Additionally, the incorporation of open-vocabulary representations within the scene graph could yield a general scene representation as opposed to solely textual node descriptions. Lastly, a potential limitation of the current system lies in the scene graph simulator and its ability to capture the various planning failures within the environment. While this works well in the cases presented in this paper, for more complex tasks involving a diverse set of predicates and affordances, the incorporation of relevant feedback messages for each instance may become infeasible and forms an important avenue for future work in this area.

## 7 Conclusion

SayPlan is a natural language-driven planning framework for robotics that integrates hierarchical 3D scene graphs and LLMs to plan across large-scale environments spanning multiple floors and rooms. We ensure the scalability of our approach by exploiting the hierarchical nature of 3D scene graphs and the semantic reasoning capabilities of LLMs to enable the agent to explore the scene graph from the highest level within the hierarchy, resulting in a significant reduction in the initial tokens required to capture larger environments. Once explored, the LLM generates task plans for a mobile manipulator robot, and a scene graph simulator ensures that the plan is feasible and grounded to the environment via iterative replanning. The framework surpasses existing techniques in producing correct, executable plans, which a robot can then follow. Finally, we successfully translate validated plans to a real-world mobile manipulator agent which operates across multiple rooms, assets and objects in a large office environment. SayPlan represents a step forward for general-purpose service robotics that can operate in our homes, hospitals and workplaces, laying the groundwork for future research in this field.

**Acknowledgments**

The authors would like to thank Ben Burgess-Limerick for assistance with the robot hardware setup, Nishant Rana for creating the illustrations and Norman Di Palo and Michael Milford for insightful discussions and feedback towards this manuscript. The authors also acknowledge the ongoing support from the QUT Centre for Robotics. This work was partially supported by the Australian Government through the Australian Research Council's Discovery Projects funding scheme (Project DP220102398) and by an Amazon Research Award to Niko Sünderhauf.

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

## A    Implementation Details

We utilise GPT-4 [3] as the underlying LLM agent unless otherwise stated. We follow a similar prompting structure to Wake et al. [5] as shown in Appendix J. We define the agent's role, details pertaining to the scene graph environment, the desired output structure and a set of input-output examples which together form the static prompt used for in-context learning. This static prompt is both task- and environment-agnostic and takes up ≈3900 tokens of the LLM's input. During semantic search, both the **3D Scene Graph** and **Memory** components of the input prompt get updated at each step, while during iterative replanning only the **Feedback** component gets updated with information from the scene graph simulator. In all cases, the LLM is prompted to output a JSON object containing arguments to call the provided API functions.

## B    Environments

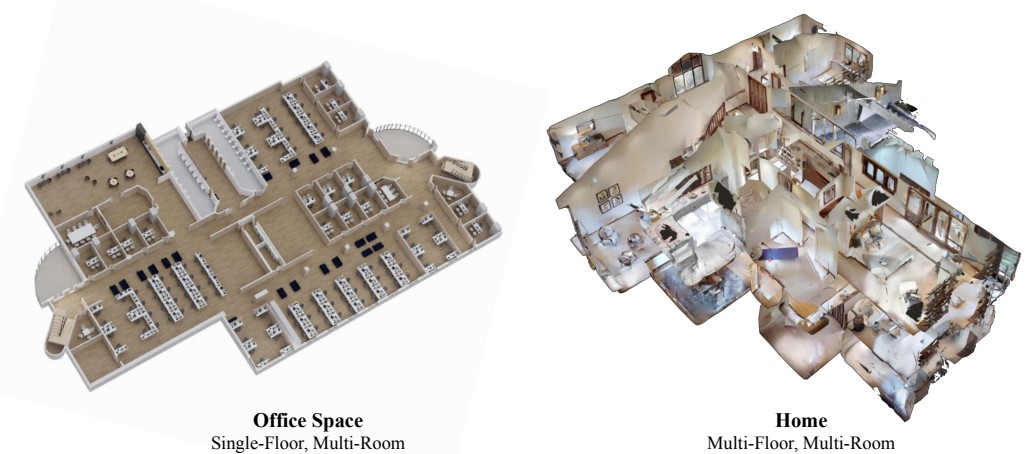

**Office Space**
Single-Floor, Multi-Room

**Home**
Multi-Floor, Multi-Room

Figure 4: **Large-scale environments used to evaluate SayPlan.** The environments span multiple rooms and floors including a vast range of

We evaluate SayPlan across a set of two large-scale environments spanning multiple rooms and floors as shown in Figure 4. We provide details of each of these environments below, including a breakdown of the number of entities and tokens required to represent them in the 3DSG:

**Office:** A large-scale office floor, spanning 37 rooms and 151 assets and objects which the agent can interact with. A full and collapsed 3D scene graph representation of this environment are provided in Appendix D and E respectively. This scene graph represents a real-world office floor within which a mobile manipulator robot is present. This allows us to embody the plans generated using SayPlan and evaluate their feasibility in the corresponding environment. Real-world video demonstrations of a mobile manipulator robot executing the generated plan in this office environment are provided on our project site[2].

**Home:** An existing 3D scene graph from the Stanford 3D Scene Graph dataset [11] which consists of a family home environment (Klickitat) spanning 28 rooms across 3 floors and contains 112 assets and objects that the agent can interact with. A 3D visual of this environment can be viewed at the 3D Scene Graph project website[3].

### B.1    Real World Environment Plan Execution

To enable real-world execution of the task plans generated over a 3DSG, we require a corresponding 2D metric map within which we can align the posed nodes captured by the 3DSG. At each room node we assume the real robot can visually locate the appropriate assets and objects that are visible to

---

[2]sayplan.github.io
[3]3dscenegraph.stanford.edu/Klickitat

| Entity Type | Number of Entities | Total Number of Tokens | Average Number of Tokens |
|---|---|---|---|
| **Room Node** | 37 | 340 | 9.19 |
| **Asset Node** | 73 | 1994 | 27.3 |
| **Object Node** | 78 | 2539 | 32.6 |
| **Agent Node** | 1 | 15 | 15.0 |
| **Node Edges** | 218 | 1843 | 8.45 |
| **Full Graph** | 407 | 6731 | 16.5 |
| **Collapsed Graph** | 105 | 878 | 8.36 |

Table 4: **Detailed 3DSG breakdown for the Office Environment.** The table summarises the number of different entities present in the 3DSG, the total LLM tokens required to represent each entity group and the average number of tokens required to represent a single type of entity.

| Entity Type | Number of Entities | Total Number of Tokens | Average Number of Tokens |
|---|---|---|---|
| **Room Node** | 28 | 231 | 8.25 |
| **Asset Node** | 52 | 1887 | 36.3 |
| **Object Node** | 60 | 1881 | 31.35 |
| **Agent Node** | 1 | 15 | 15 |
| **Node Edges** | 323 | 2584 | 8 |
| **Full Graph** | 464 | 6598 | 14.2 |
| **Collapsed Graph** | 240 | 1817 | 7.57 |

Table 5: **Detailed 3DSG breakdown for the Home Environment.** The table summarises the number of different entities present in the 3DSG, the total LLM tokens required to represent each entity group and the average number of tokens required to represent a single type of entity.

it within the 3DSG. The mobile manipulator robot used for the demonstration consisted of a Franka Panda 7-DoF robot manipulator [47] attached to an LD-60 Omron mobile base [48]. The robot is equipped with a LiDAR scanner to localise the robot both within the real world and the corresponding 3DSG. All the skills or affordances including pick, place, open and close were developed using the motion controller from [44] coupled with a RGB-D vision module for grasp detection, and a behaviour tree to manage the execution of each component including failure recovery. Future work could incorporate a range of pre-trained skills (whisking, flipping, spreading etc.) using imitation learning [49, 50] or reinforcement learning [51, 52] to increase the diversity of tasks that SayPlan is able to achieve.

## C   Tasks

| Instruction Family | Num | Explanation | Example Instruction |
|---|---|---|---|
| | | **Semantic Search** | |
| Simple Search | 30 | Queries focussed on evaluating the basic semantic search capabilities of SayPlan | Find me a ripe banana. |
| Complex Search | 30 | Abstract semantic search queries which require complex reasoning | Find the room where people are playing board games. |
| | | **Causal Planning** | |
| Simple Planning | 15 | Queries which require the agent to perform search, causal reasoning and environment interaction in order to solve a task. | Refrigerate the orange left on the kitchen bench. |
| Long-Horizon Planning | 15 | Long Horizon planning queries requiring multiple interactive steps | Tobi spilt soda on his desk. Help him clean up. |

Table 6: **List of evaluation task instructions.** We evaluate SayPlan on 90 instructions, grouped to test various aspects of the planning capabilities across large-scale scene graphs. The full instruction set is given in Appendix C.

We evaluate SayPlan across 4 instruction sets which are classified to evaluate different aspects of its 3D scene graph reasoning and planning capabilities as shown in Table 6:

**Simple Search:** Focused on evaluating the semantic search capabilities of the LLM based on queries which directly reference information in the scene graph as well as the basic graph-based reasoning capabilities of the LMM.

**Complex Search:** Abstract semantic search queries which require complex reasoning. The information required to solve these search tasks is not readily available in the graph and has to be inferred by the underlying LLM.

**Simple Planning:** Task planning queries which require the agent to perform graph search, causal reasoning and environment interaction in order to solve the task. Typically requires shorter horizon plans over single rooms.

**Long Horizon Planning:** Long Horizon planning queries require multiple interactive steps. These queries evaluate SayPlan's ability to reason over temporally extended instructions to investigate how well it scales to such regimes. Typically requires long horizon plans spanning multiple rooms.

The full list of instructions used and the corresponding aspect the query evaluates are given in the following tables:

## C.1 Simple Search

### C.1.1 Office Environment

| Instruction | |
|---|---|
| Find me object K31X. | ▷ unguided search with no semantic cue |
| Find me a carrot. | ▷ semantic search based on node name |
| Find me anything purple in the postdoc bays. | ▷ semantic search with termination conditioned on attribute |
| Find me a ripe banana. | ▷ semantic search with termination conditioned on attribute |
| Find me something that has a screwdriver in it. | ▷ unguided search with termination conditioned on children |
| One of the offices has a poster of the Terminator. Which one is it? | ▷ semantic search with termination conditioned on children |
| I printed a document but I don't know which printer has it. Find the document. | ▷ semantic search based on parent |
| I left my headphones in one of the meeting rooms. Locate them. | ▷ semantic search based on parent |
| Find the PhD bay that has a drone in it. | ▷ semantic search with termination conditioned on children |
| Find the kale that is not in the kitchen. | ▷ semantic search with termination conditioned on a negation predicate on parent |
| Find me an office that does not have a cabinet. | ▷ semantic search with termination conditioned on a negation predicate on children |
| Find me an office that contains a cabinet, a desk, and a chair. | ▷ semantic search with termination conditioned on a conjunctive query on children |
| Find a book that was left next to a robotic gripper. | ▷ semantic search with termination conditioned on a sibling |
| Luis gave one of his neighbours a stapler. Find the stapler. | ▷ semantic search with termination conditioned on a sibling |
| There is a meeting room with a chair but no table. Locate it. | ▷ semantic search with termination conditioned on a conjunctive query with negation |

Table 7: **Simple Search Instructions.** Evaluated in Office Environment.

### C.1.2 Home Environment

| Instruction | |
|---|---|
| Find me a FooBar. | ▷ unguided search with no semantic cue |
| Find me a bottle of wine. | ▷ semantic search based on node name |
| Find me a plant with thorns. | ▷ semantic search with termination conditioned on attribute |
| Find me a plant that needs watering. | ▷ semantic search with termination conditioned on attribute |
| Find me a bathroom with no toilet. | ▷ semantic search with termination conditioned on a negation predicate |
| The baby dropped their rattle in one of the rooms. Locate it. | ▷ semantic search based on node name |
| I left my suitcase either in the bedroom or the living room. Which room is it in. | ▷ semantic search based on node name |
| Find the room with a ball in it. | ▷ semantic search based on node name |
| I forgot my book on a bed. Locate it. | ▷ semantic search based on node name |
| Find an empty vase that was left next to sink. | ▷ semantic search with termination conditioned on sibling |
| Locate the dining room which has a table, chair and a baby monitor. | ▷ semantic search with termination conditioned on conjuctive query |
| Locate a chair that is not in any dining room. | ▷ semantic search with termination conditioned on negation predicate |
| I need to shave. Which room has both a razor and shaving cream. | ▷ semantic search with termination conditioned on children |
| Find me 2 bedrooms with pillows in them. | ▷ semantic search with multiple returns |
| Find me 2 bedrooms without pillows in them. | ▷ semantic search with multiple returns based on negation predicate |

Table 8: **Simple Search Instructions.** Evaluated in Home Environment.

### C.2 Complex Search

### C.2.1 Office Environment

| Instruction | |
|---|---|
| Find object J64M. J64M should be kept at below 0 degrees Celsius. | ▷ semantic search guided by implicit world knowledge (knowledge not directly encoded in graph) |
| Find me something non vegetarian. | ▷ semantic search with termination conditioned on implicit world knowledge |
| Locate something sharp. | ▷ unguided search with termination conditioned on implicit world knowledge |
| Find the room where people are playing board games. | ▷ semantic search with termination conditioned on ability to deduce context from node children using world knowledge ("board game" is not part of any node name or attribute in this graph) |
| Find an office of someone who is clearly a fan of Arnold Schwarzenegger. | ▷ semantic search with termination conditioned on ability to deduce context from node children using world knowledge |
| There is a postdoc that has a pet Husky. Find the desk that's most likely theirs. | ▷ semantic search with termination conditioned on ability to deduce context from node children using world knowledge |
| One of the PhD students was given more than one complimentary T-shirts. Find his desk. | ▷ semantic search with termination conditioned on the number of children |
| Find me the office where a paper attachment device is inside an asset that is open. | ▷ semantic search with termination conditioned on node descendants and their attributes |
| There is an office which has a cabinet containing exactly 3 items in it. Locate the office. | ▷ semantic search with termination conditioned on the number of children |
| There is an office which has a cabinet containing a rotten apple. The cabinet name contains an even number. Locate the office. | ▷ semantic search guided by numerical properties |
| Look for a carrot. The carrot is likely to be in a meeting room but I'm not sure. | ▷ semantic search guided by user provided bias |
| Find me a meeting room with a RealSense camera. | ▷ semantic search that has no result (no meeting room has a realsense camera in the graph) |
| Find the closest fire extinguisher to the manipulation lab. | ▷ search guided by node distance |
| Find me the closest meeting room to the kitchen. | ▷ search guided by node distance |
| Either Filipe or Tobi has my headphones. Locate it. | ▷ evaluating constrained search, early termination once the two office are explored |

Table 9: **Complex Search Instructions.** Evaluated in Office Environment.

| Instruction | |
|---|---|
| I need something to access ChatGPT. Where should I go? | ▷ semantic search guided by implicit world knowledge |
| Find the livingroom that contains the most electronic devices. | ▷ semantic search with termination conditioned on children with indirect information |
| Find me something to eat with a lot of potassium. | ▷ semantic search with termination conditioned on implicit world knowledge |
| I left a sock in a bedroom and one in the living room. Locate them. They should match. | ▷ semantic search with multiple returns |
| Find me a potted plant that is most likely a cactus. | ▷ semantic search with termination implicitly conditioned on attribute |
| Find the dining room with exactly 5 chairs. | ▷ semantic search with termination implicitly conditioned on quantity of children |
| Find me the bedroom closest to the home office. | ▷ semantic search with termination implicitly conditioned on node distance |
| Find me a bedroom with an unusual amount of bowls. | ▷ semantic search with termination implicitly conditioned on quantity of children |
| Which bedroom is empty. | ▷ semantic search with termination implicitly conditioned on quantity of children |
| Which bathroom has the most potted plants. | ▷ semantic search with termination implicitly conditioned on quantity of children |
| The kitchen is flooded. Find somewhere I can heat up my food. | ▷ semantic search guided by negation |
| Find me the room which most likely belongs to a child | ▷ semantic search with termination conditioned on ability to deduce context from node children using world knowledge |
| 15 guests are arriving. Locate enough chairs to seat them. | ▷ semantic search with termination implicitly conditioned on the quantity of specified node |
| A vegetarian dinner was prepared in one of the dining rooms. Locate it. | ▷ semantic search with selection criteria based on world knowledge |
| My tie is in one of the closets. Locate it. | ▷ evaluating constrained search that has no result, termination after exploring closets |

Table 10: **Complex Search Instructions.** Evaluated in Home Environment.

## C.3 Simple Planning

| Instruction |
| --- |
| Close Jason's cabinet. |
| Refrigerate the orange left on the kitchen bench. |
| Take care of the dirty plate in the lunchroom. |
| Place the printed document on Will's desk. |
| Peter is working hard at his desk. Get him a healthy snack. |
| Hide one of Peter's valuable belongings. |
| Wipe the dusty admin shelf. |
| There is coffee dripping on the floor. Stop it. |
| Place Will's drone on his desk. |
| Move the monitor from Jason's office to Filipe's. |
| My parcel just got delivered! Locate it and place it in the appropriate lab. |
| Check if the coffee machine is working. |
| Heat up the chicken kebab. |
| Something is smelling in the kitchen. Dispose of it. |
| Throw what the agent is holding in the bin. |

Table 11: **Simple Planning Instructions.** Evaluated in Office Environment.

## C.4 Long Horizon Planning

| Instruction |
| --- |
| Heat up the noodles in the fridge, and place it somewhere where I can enjoy it. |
| Throw the rotting fruit in Dimity's office in the correct bin. |
| Wash all the dishes on the lunch table. Once finished, place all the clean cutlery in the drawer. |
| Safely file away the freshly printed document in Will's office then place the undergraduate thesis on his desk. |
| Make Niko a coffee and place the mug on his desk. |
| Someone has thrown items in the wrong bins. Correct this. |
| Tobi spilt soda on his desk. Throw away the can and take him something to clean with. |
| I want to make a sandwich. Place all the ingredients on the lunch table. |
| A delegation of project partners is arriving soon. We want to serve them snacks and non-alcoholic drinks. Prepare everything in the largest meeting room. Use items found in the supplies room only. |
| Serve bottled water to the attendees who are seated in meeting room 1. Each attendee can only receive a single bottle of water. |
| Empty the dishwasher. Place all items in their correct locations |
| Locate all 6 complimentary t-shirts given to the PhD students and place them on the shelf in admin. |
| I'm hungry. Bring me an apple from Peter and a pepsi from Tobi. I'm at the lunch table. |
| Let's play a prank on Niko. Dimity might have something. |
| There is an office which has a cabinet containing a rotten apple. The cabinet name contains an even number. Locate the office, throw away the fruit and get them a fresh apple. |

Table 12: **Long-Horizon Planning Instructions.** Evaluated in Office Environment.

# D   Full 3D Scene Graph: Office Environment

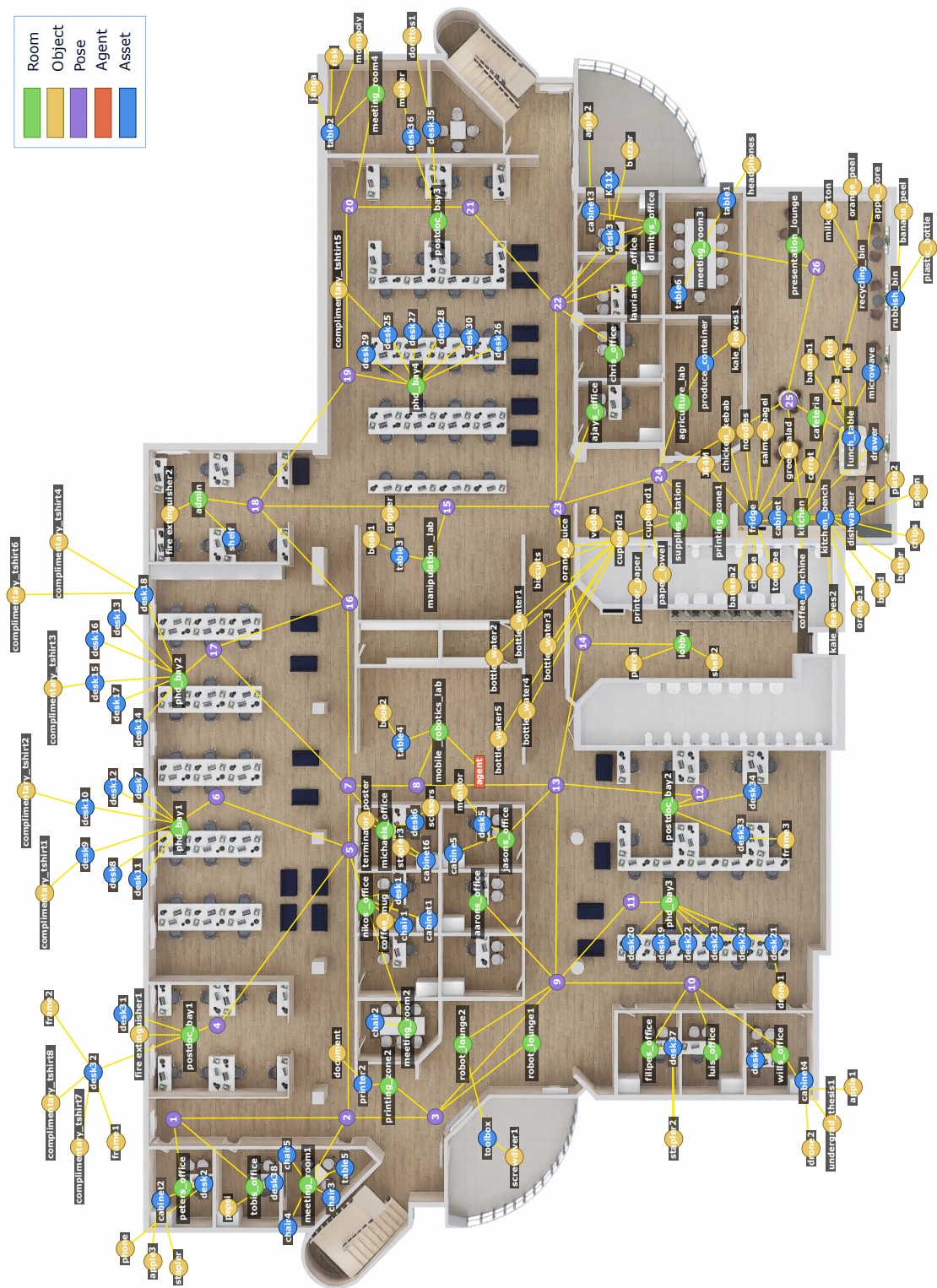

Figure 5: **3D Scene Graph - Fully Expanded Office Environment.** Full 3D scene graph exposing all the rooms, assets and objects available in the scene. Note that the LLM agent never sees all this information unless it chooses to expand every possible node without contraction.

# E Contracted 3D Scene Graph: Office Environment

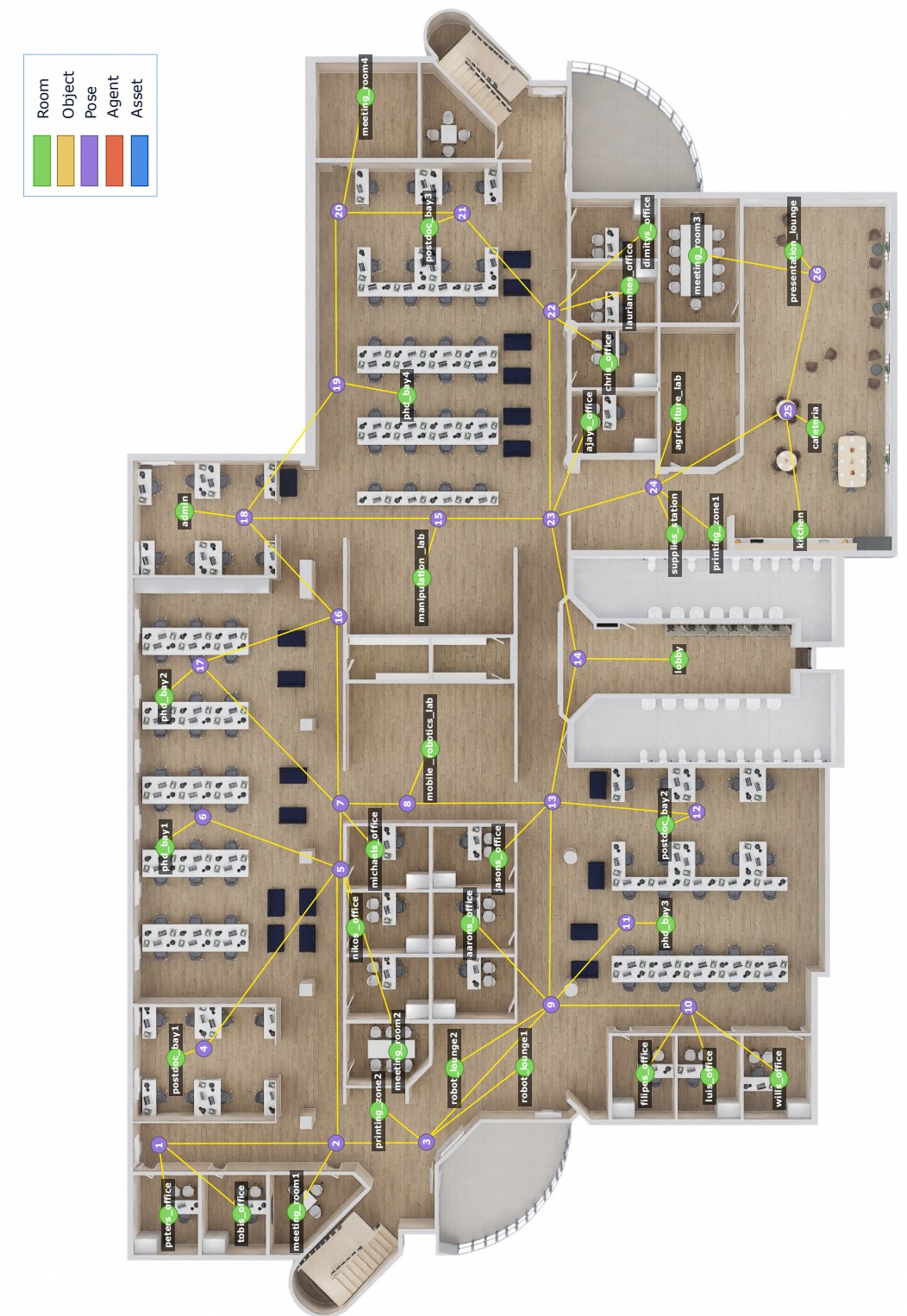

Figure 6: **3D Scene Graph - Contracted Office Environment.** Contracted 3D scene graph exposing only the highest level within the hierarchy - room nodes. This results in an 82.1% reduction in the number of tokens required to represent the scene before the semantic search phase.

# F  Semantic Search Evaluation Results

- Full listings of the generated semantic search sequences for the evaluation instruction sets are provided on the following pages -

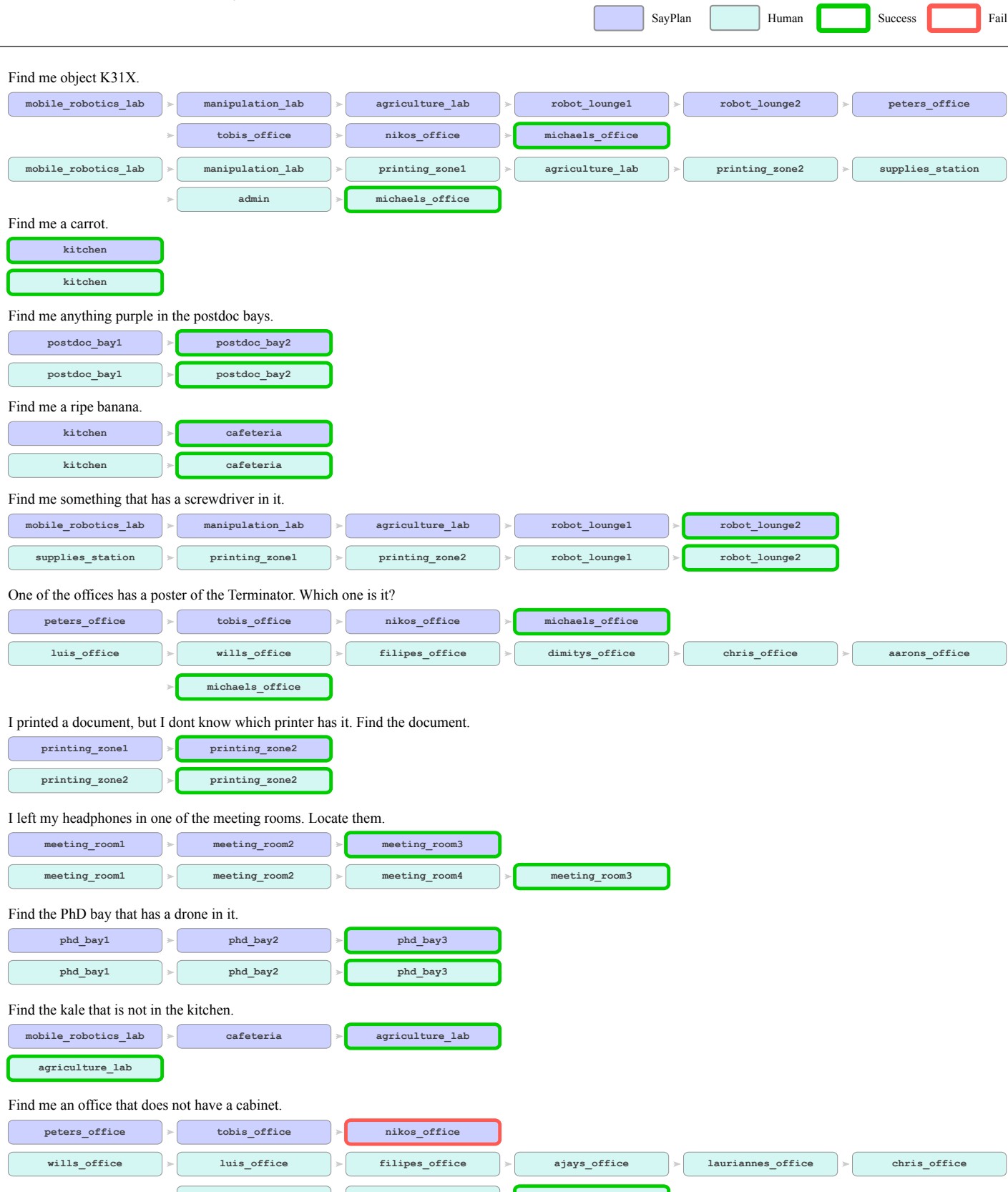

Find me object K31X.

| mobile_robotics_lab | manipulation_lab | agriculture_lab | robot_lounge1 | robot_lounge2 | peters_office |
| tobis_office | nikos_office | **michaels_office** |
| mobile_robotics_lab | manipulation_lab | printing_zone1 | agriculture_lab | printing_zone2 | supplies_station |
| admin | **michaels_office** |

Find me a carrot.

**kitchen**

**kitchen**

Find me anything purple in the postdoc bays.

| postdoc_bay1 | **postdoc_bay2** |
| postdoc_bay1 | **postdoc_bay2** |

Find me a ripe banana.

| kitchen | **cafeteria** |
| kitchen | **cafeteria** |

Find me something that has a screwdriver in it.

| mobile_robotics_lab | manipulation_lab | agriculture_lab | robot_lounge1 | **robot_lounge2** |
| supplies_station | printing_zone1 | printing_zone2 | robot_lounge1 | **robot_lounge2** |

One of the offices has a poster of the Terminator. Which one is it?

| peters_office | tobis_office | nikos_office | **michaels_office** |
| luis_office | wills_office | filipes_office | dimitys_office | chris_office | aarons_office |
| **michaels_office** |

I printed a document, but I dont know which printer has it. Find the document.

| printing_zone1 | **printing_zone2** |
| printing_zone2 | **printing_zone2** |

I left my headphones in one of the meeting rooms. Locate them.

| meeting_room1 | meeting_room2 | **meeting_room3** |
| meeting_room1 | meeting_room2 | meeting_room4 | **meeting_room3** |

Find the PhD bay that has a drone in it.

| phd_bay1 | phd_bay2 | **phd_bay3** |
| phd_bay1 | phd_bay2 | **phd_bay3** |

Find the kale that is not in the kitchen.

| mobile_robotics_lab | cafeteria | **agriculture_lab** |
| **agriculture_lab** |

Find me an office that does not have a cabinet.

| peters_office | tobis_office | nikos_office |
| wills_office | luis_office | filipes_office | ajays_office | lauriannes_office | chris_office |
| dimitys_office | peters_office | **tobis_office** |

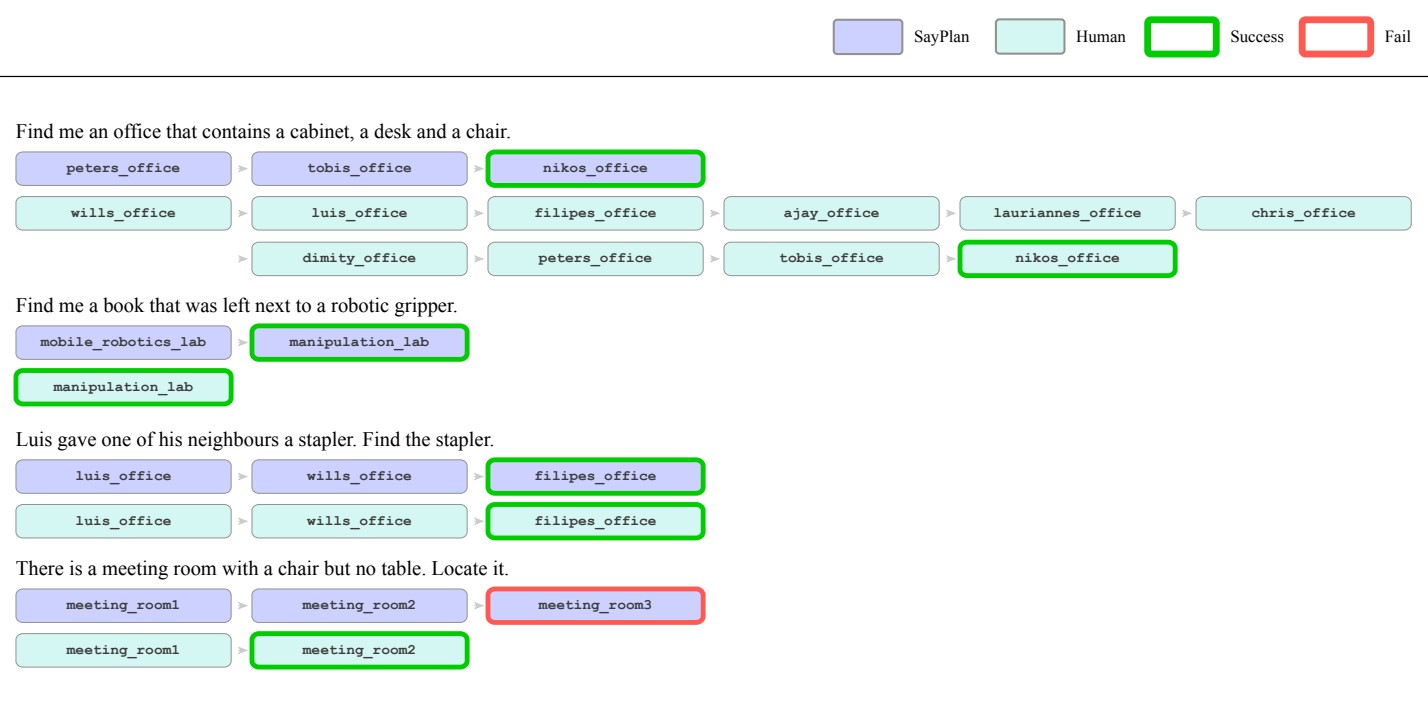

Find me an office that contains a cabinet, a desk and a chair.

peters_office → tobis_office → nikos_office

wills_office → luis_office → filipes_office → ajay_office → lauriannes_office → chris_office

→ dimity_office → peters_office → tobis_office → nikos_office

Find me a book that was left next to a robotic gripper.

mobile_robotics_lab → manipulation_lab

manipulation_lab

Luis gave one of his neighbours a stapler. Find the stapler.

luis_office → wills_office → filipes_office

luis_office → wills_office → filipes_office

There is a meeting room with a chair but no table. Locate it.

meeting_room1 → meeting_room2 → meeting_room3

meeting_room1 → meeting_room2

Table 13: **Simple Search Office Environment Evaluation.** Sequence of Explored Nodes for Simple Search Office Environment Instructions.

Find object J64M. J64M should be kept at below 0 degrees Celsius.

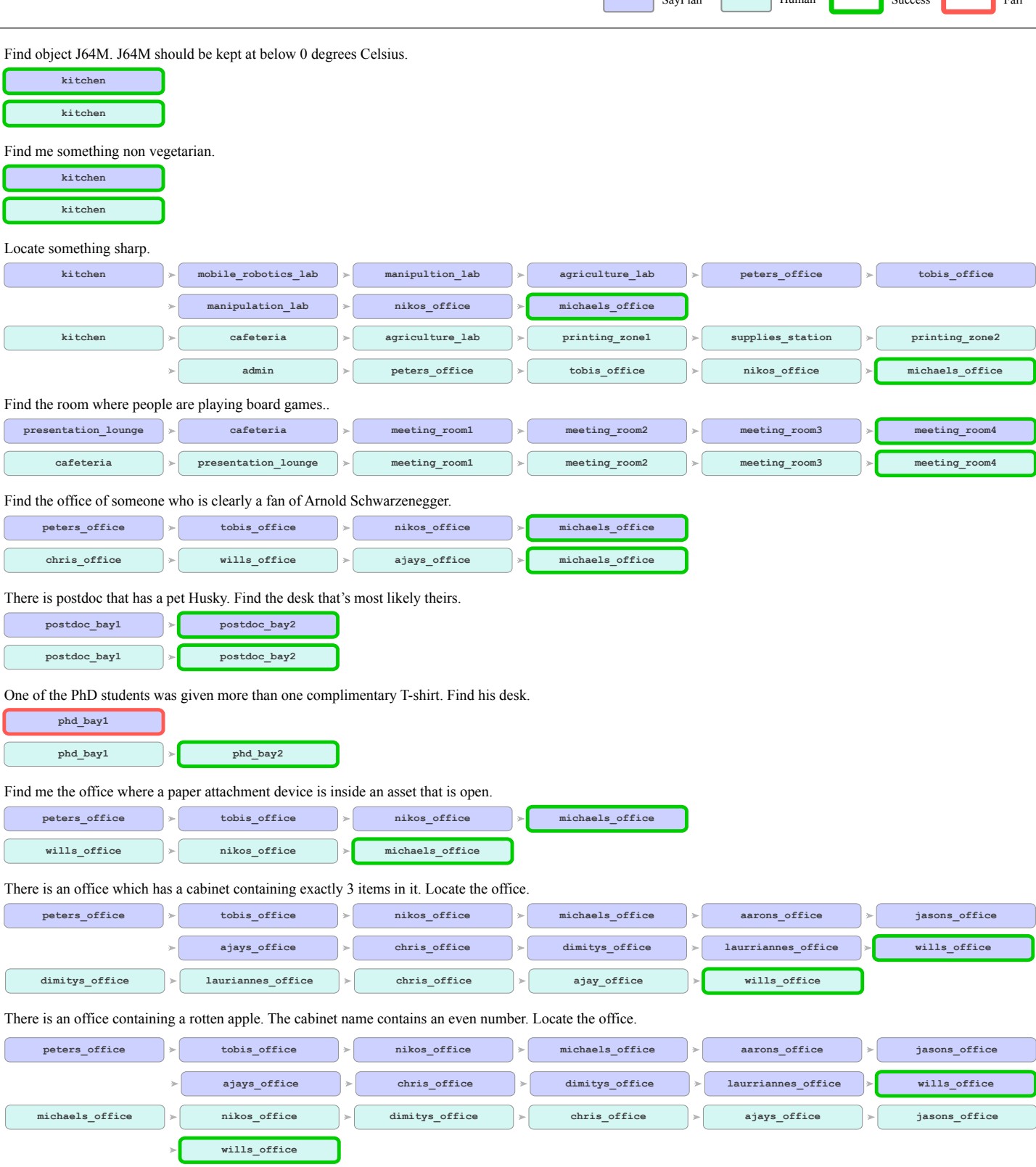

Find me something non vegetarian.

Locate something sharp.

Find the room where people are playing board games..

Find the office of someone who is clearly a fan of Arnold Schwarzenegger.

There is postdoc that has a pet Husky. Find the desk that's most likely theirs.

One of the PhD students was given more than one complimentary T-shirt. Find his desk.

Find me the office where a paper attachment device is inside an asset that is open.

There is an office which has a cabinet containing exactly 3 items in it. Locate the office.

There is an office containing a rotten apple. The cabinet name contains an even number. Locate the office.

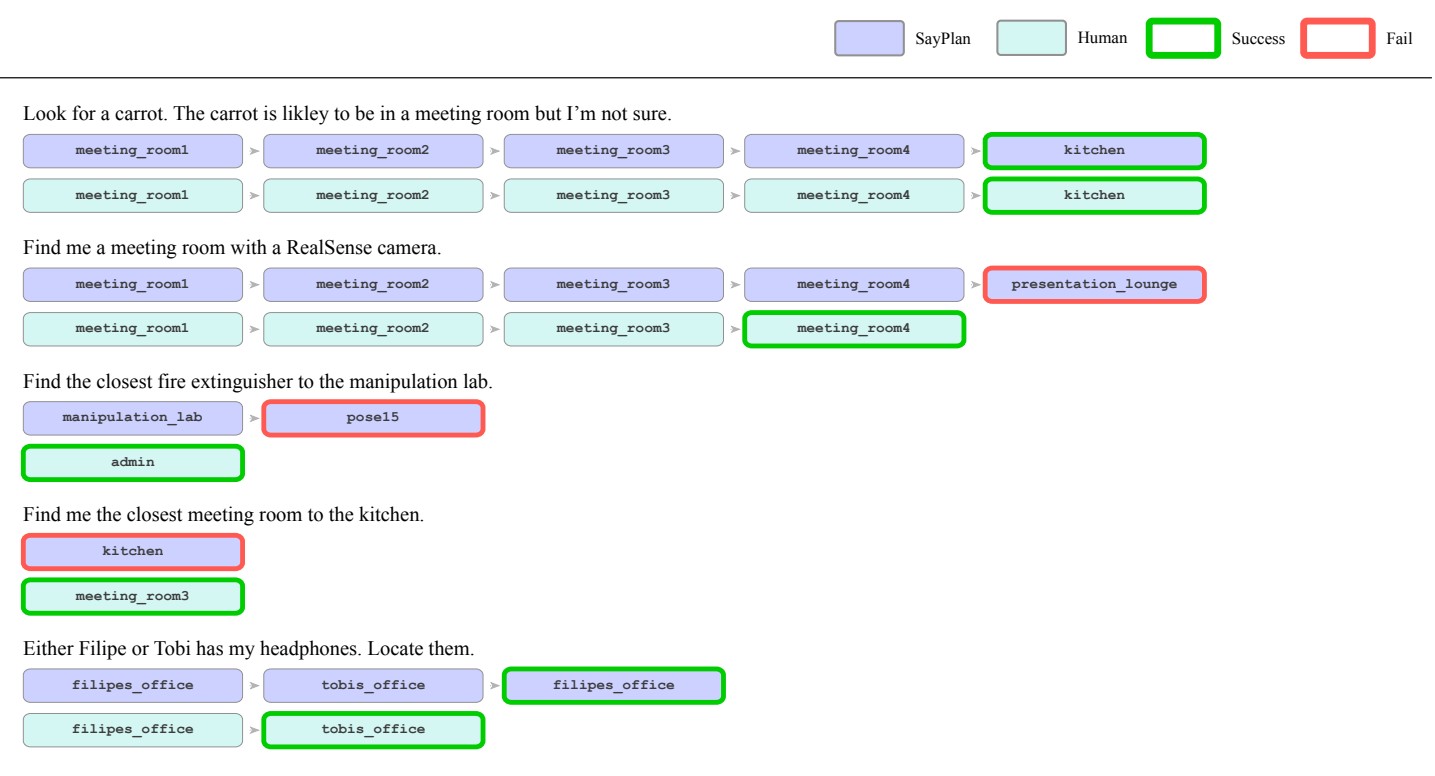

Table 14: **Complex Search Office Environment Evaluation.** Sequence of Explored Nodes for Complex Search Office Environment Instructions.

**Find me a FooBar.**

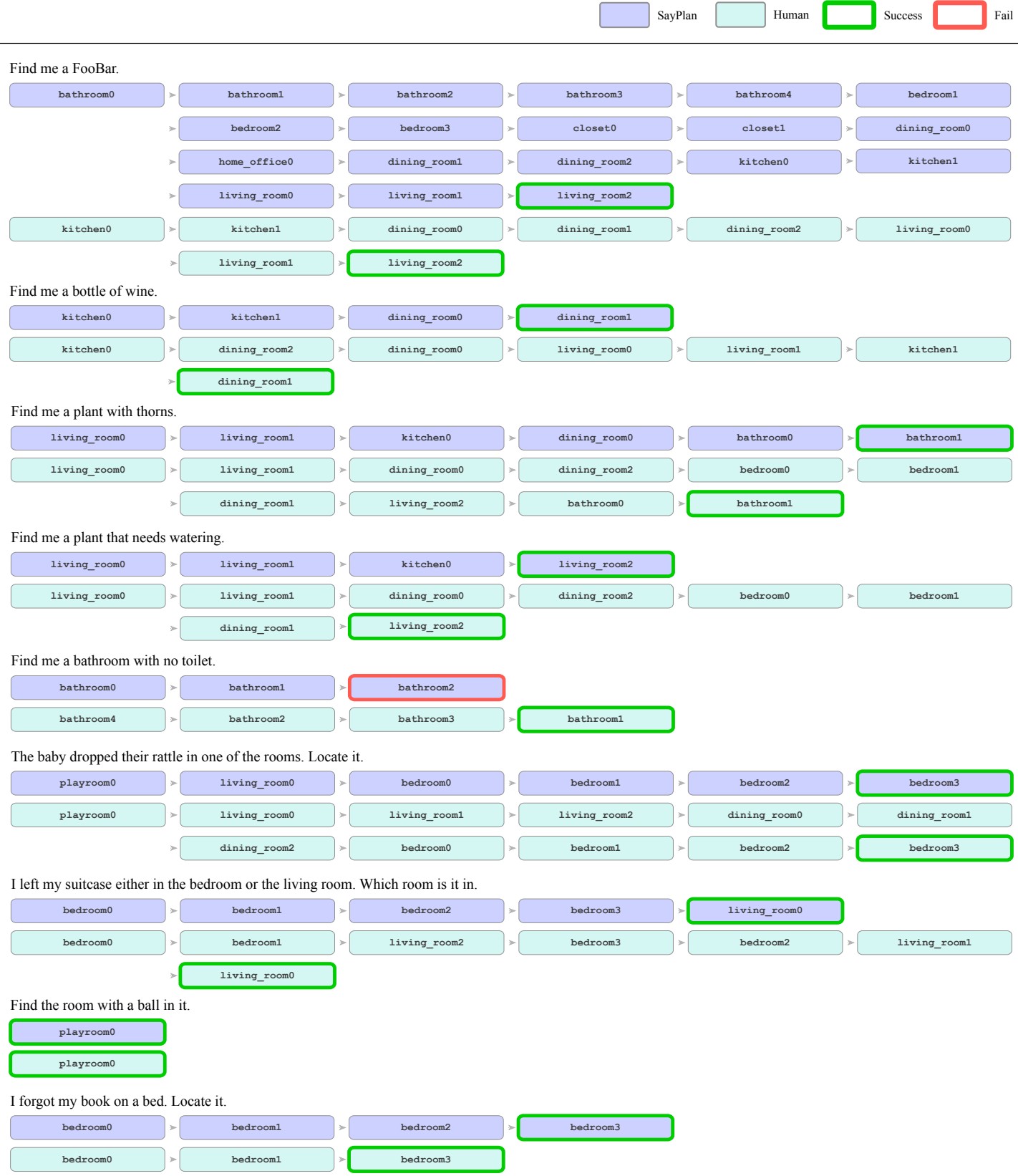

**Find me a bottle of wine.**

**Find me a plant with thorns.**

**Find me a plant that needs watering.**

**Find me a bathroom with no toilet.**

**The baby dropped their rattle in one of the rooms. Locate it.**

**I left my suitcase either in the bedroom or the living room. Which room is it in.**

**Find the room with a ball in it.**

**I forgot my book on a bed. Locate it.**

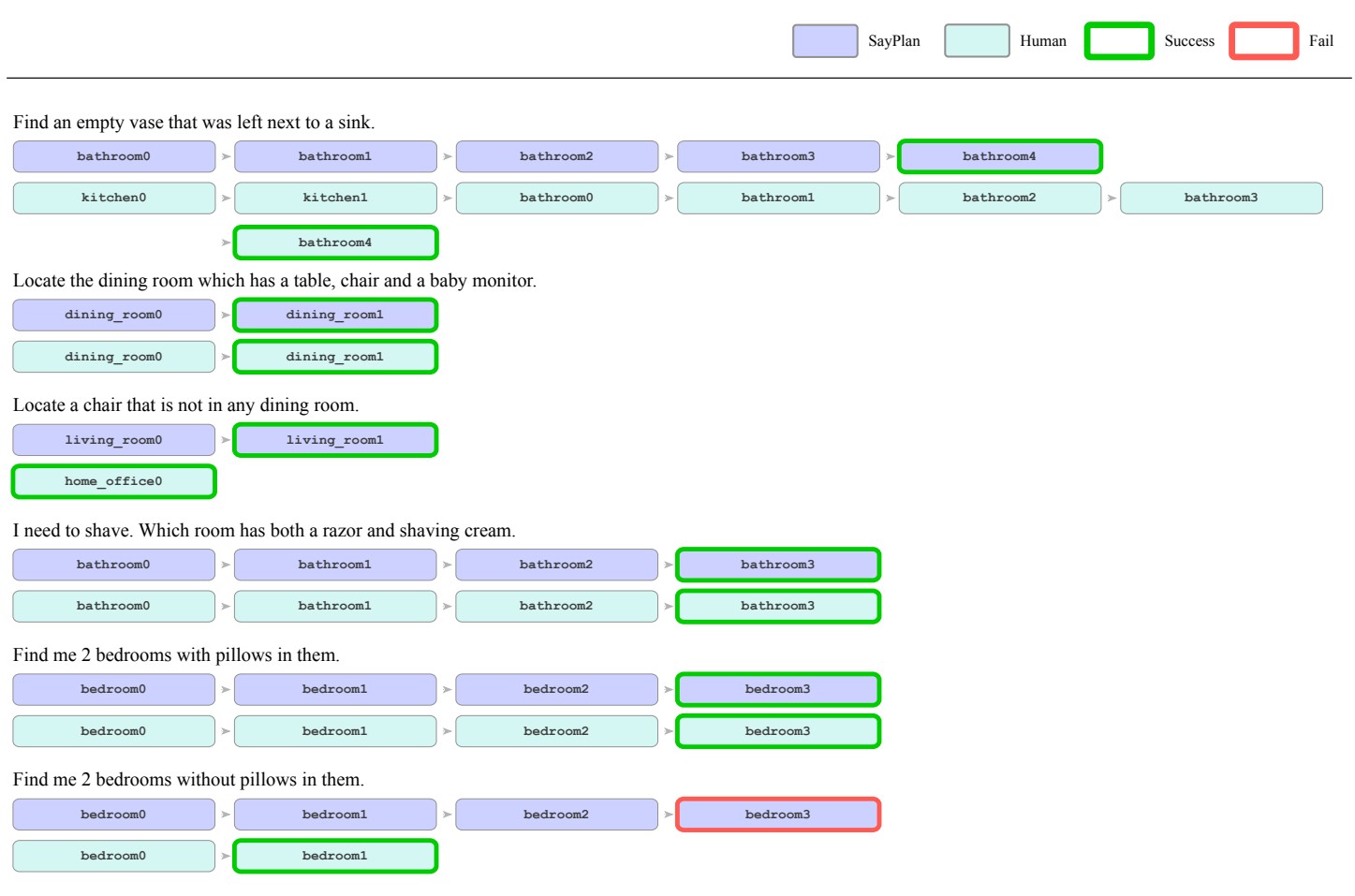

Table 15: **Simple Search Home Environment Evaluation.** Sequence of Explored Nodes for Simple Search Home Environment Instructions.

I need something to access ChatGPT. Where should I go?.

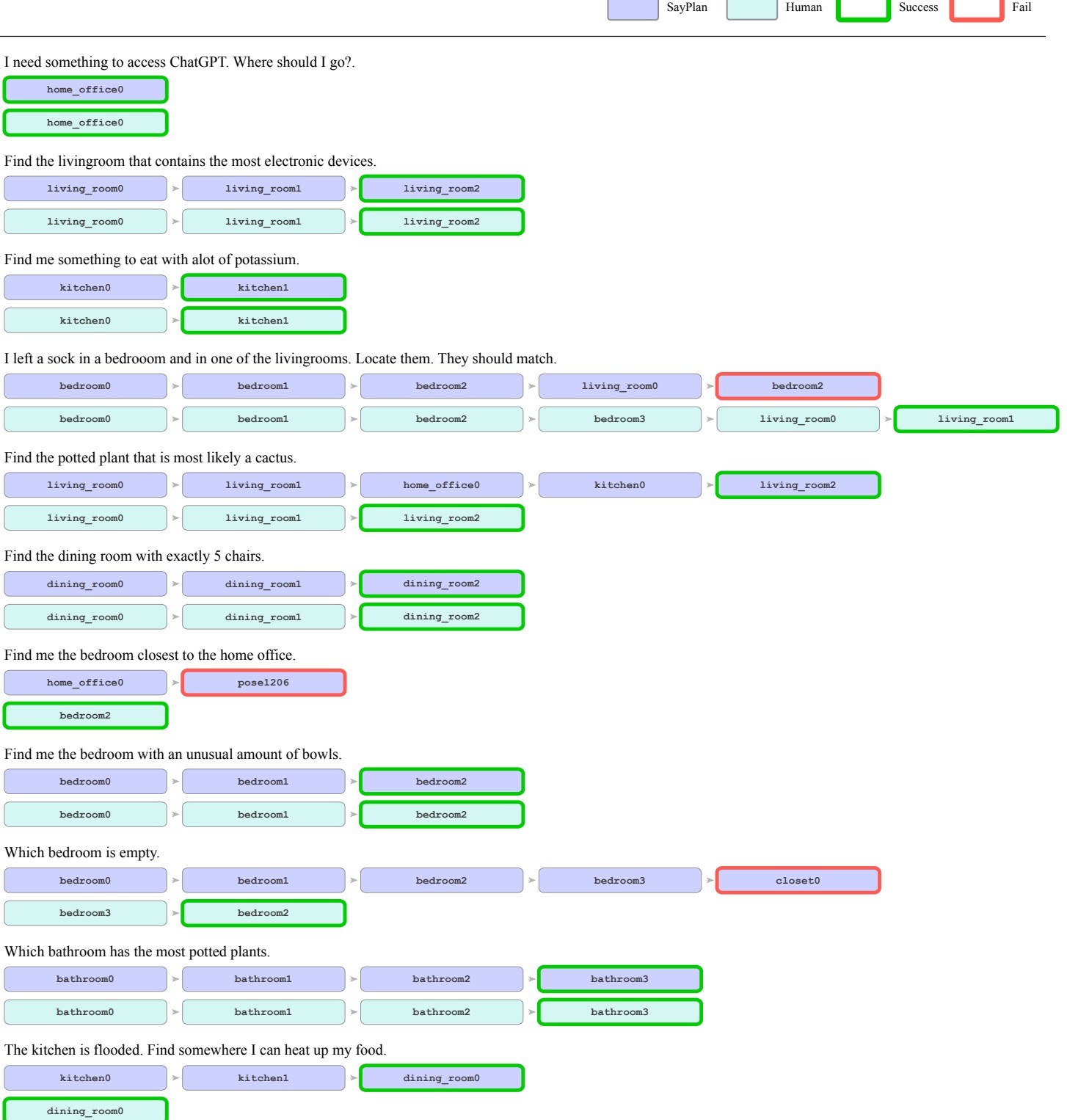

Find the livingroom that contains the most electronic devices.

Find me something to eat with alot of potassium.

I left a sock in a bedrooom and in one of the livingrooms. Locate them. They should match.

Find the potted plant that is most likely a cactus.

Find the dining room with exactly 5 chairs.

Find me the bedroom closest to the home office.

Find me the bedroom with an unusual amount of bowls.

Which bedroom is empty.

Which bathroom has the most potted plants.

The kitchen is flooded. Find somewhere I can heat up my food.

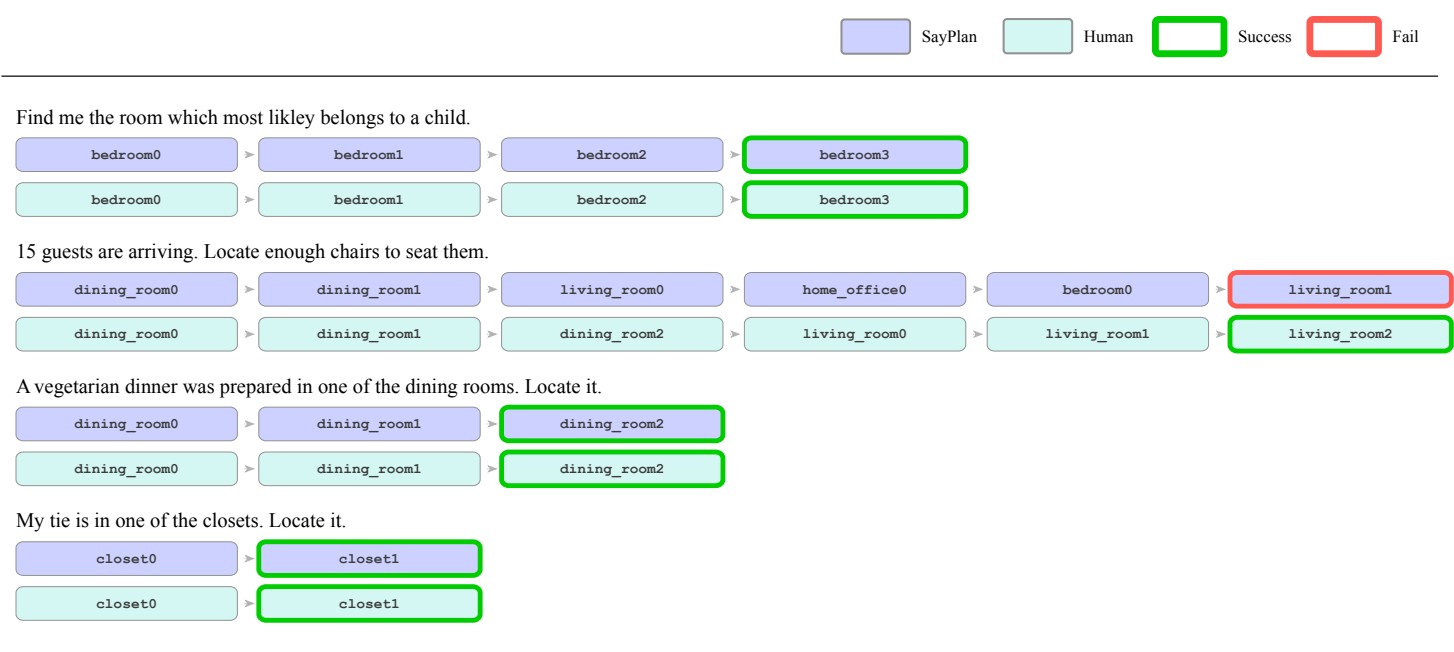

Table 16: **Complex Search Home Environment Evaluation.** Sequence of Explored Nodes for Complex Search Home Environment Instructions.

# G Causal Planning Evaluation Results

In this section, we provide a detailed breakdown of the causal planning performance of SayPlan across the two sets of evaluation instructions. Tables 17 and 18 detail the correctness, executability and the number of iterative replanning steps it took to obtain an executable plan.

| Instruction | Corr. | Exec. | No. of Replanning Iterations |
|---|:---:|:---:|:---:|
| Close Jason's cabinet. | ✓ | ✓ | 0 |
| Refrigerate the orange left on the kitchen bench. | ✓ | ✓ | 0 |
| Take care of the dirty plate in the lunchroom. | ✓ | ✓ | 0 |
| Place the printed document on Will's desk. | ✓ | ✓ | 0 |
| Peter is working hard at his desk. Get him a healthy snack. | ✗ | ✓ | 5 |
| Hide one of Peter's valuable belongings. | ✓ | ✓ | 0 |
| Wipe the dusty admin shelf. | ✓ | ✓ | 0 |
| There is coffee dripping on the floor. Stop it. | ✓ | ✓ | 0 |
| Place Will's drone on his desk. | ✓ | ✓ | 0 |
| Move the monitor from Jason's office to Filipe's. | ✓ | ✓ | 0 |
| My parcel just got delivered! Locate it and place it in the appropriate lab. | ✓ | ✓ | 0 |
| Check if the coffee machine is working. | ✓ | ✓ | 0 |
| Heat up the chicken kebab. | ✓ | ✓ | 1 |
| Something is smelling in the kitchen. Dispose of it. | ✓ | ✓ | 0 |
| Throw what the agent is holding in the bin. | ✓ | ✓ | 1 |

Table 17: **Correctness, Executability and Number of Replanning Iterations for *Simple Planning* Instructions.** Evaluating the performance of SayPlan on each simple planning instruction. Values indicated in red indicate that no executable plan was identified up to that number of iterative replanning steps. In this case, 5 was the maximum number of replanning steps.

| Instruction | Corr. | Exec. | No. of Replanning Iterations |
|---|---|---|---|
| Heat up the noodles in the fridge, and place it somewhere where I can enjoy it. | ✓ | ✓ | 2 |
| Throw the rotting fruit in Dimity's office in the correct bin. | ✓ | ✓ | 1 |
| Wash all the dishes on the lunch table. Once finished, place all the clean cutlery in the drawer. | ✗ | ✓ | 2 |
| Safely file away the freshly printed document in Will's office then place the undergraduate thesis on his desk. | ✓ | ✓ | 2 |
| Make Niko a coffee and place the mug on his desk. | ✓ | ✓ | 0 |
| Someone has thrown items in the wrong bins. Correct this. | ✗ | ✓ | 0 |
| Tobi spilt soda on his desk. Throw away the can and take him something to clean with. | ✓ | ✓ | 3 |
| I want to make a sandwich. Place all the ingredients on the lunch table. | ✓ | ✓ | 3 |
| A delegation of project partners is arriving soon. We want to serve them snacks and non-alcoholic drinks. Prepare everything in the largest meeting room. Use items found in the supplies room only. | ✓ | ✓ | 2 |
| Serve bottled water to the attendees who are seated in meeting room 1. Each attendee can only receive a single bottle of water. | ✓ | ✓ | 2 |
| Empty the dishwasher. Place all items in their correct locations. | ✓ | ✓ | 2 |
| Locate all 6 complimentary t-shirts given to the PhD students and place them on the shelf in admin. | ✓ | ✓ | 1 |
| I'm hungry. Bring me an apple from Peter and a Pepsi from Tobi. I'm at the lunch table. | ✗ | ✗ | 5 |
| Let's play a prank on Niko. Dimity might have something. | ✓ | ✓ | 1 |
| There is an office which has a cabinet containing a rotten apple. The cabinet name contains an even number. Locate the office, throw away the fruit and get them a fresh apple. | ✗ | ✗ | 5 |

Table 18: **Correctness, Executability and Number of Replanning Iterations for *Long-Horizon Planning* Instructions.** Evaluating the performance of SayPlan on each long-horizon planning instruction. Values indicated in red indicate that no executable plan was identified up to that number of iterative replanning steps. In this case, 5 was the maximum number of replanning steps.

The full plan sequences generated by SayPlan and all the baseline methods for each of the above instructions are detailed in Table 19. Note the regions highlighted in red indicating the precise action where a plan failed.

- Full listings of the generated plan sequences for the simple and long-horizon instruction sets are provided on the following pages -

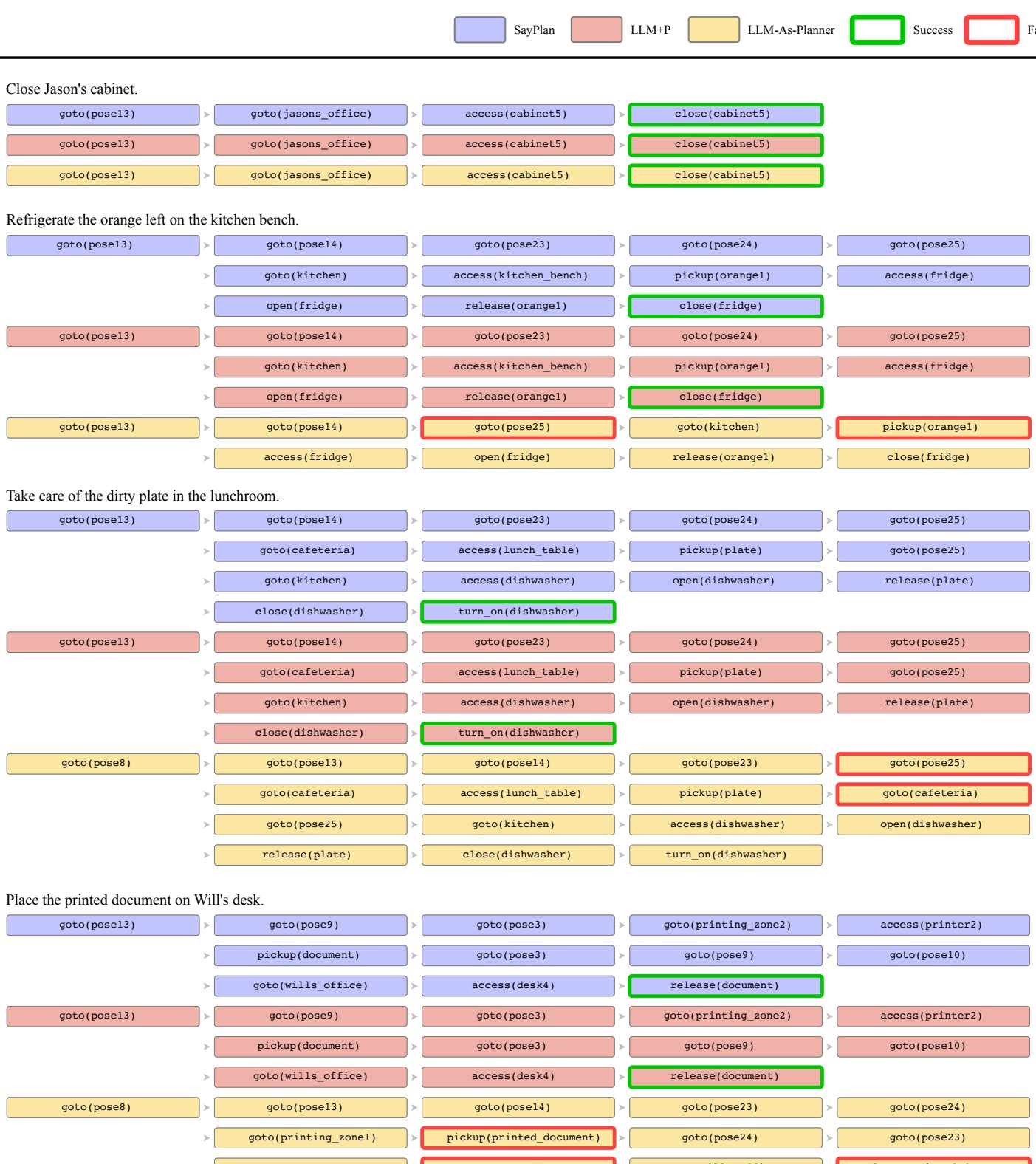

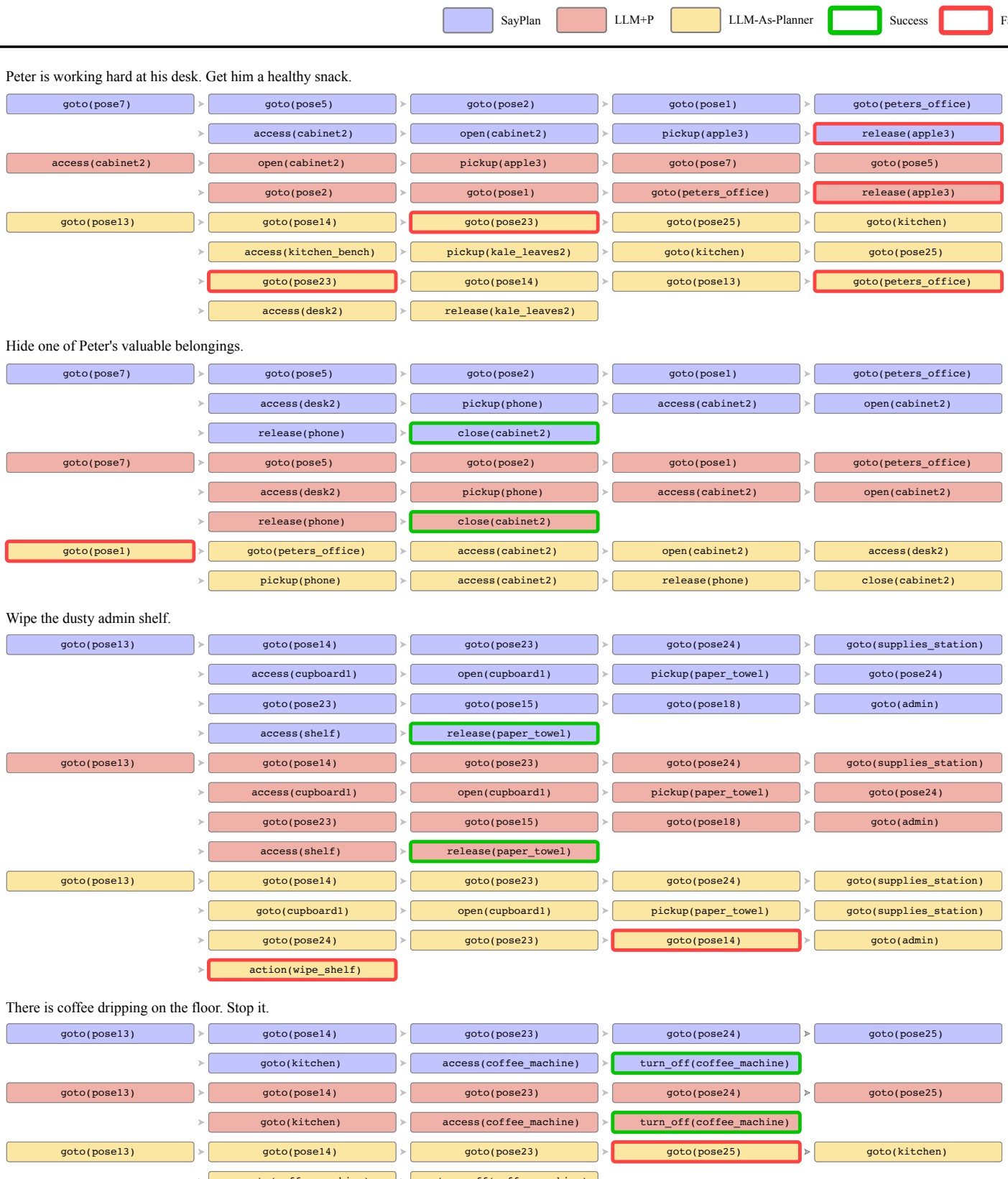

Place Will's drone on his desk.

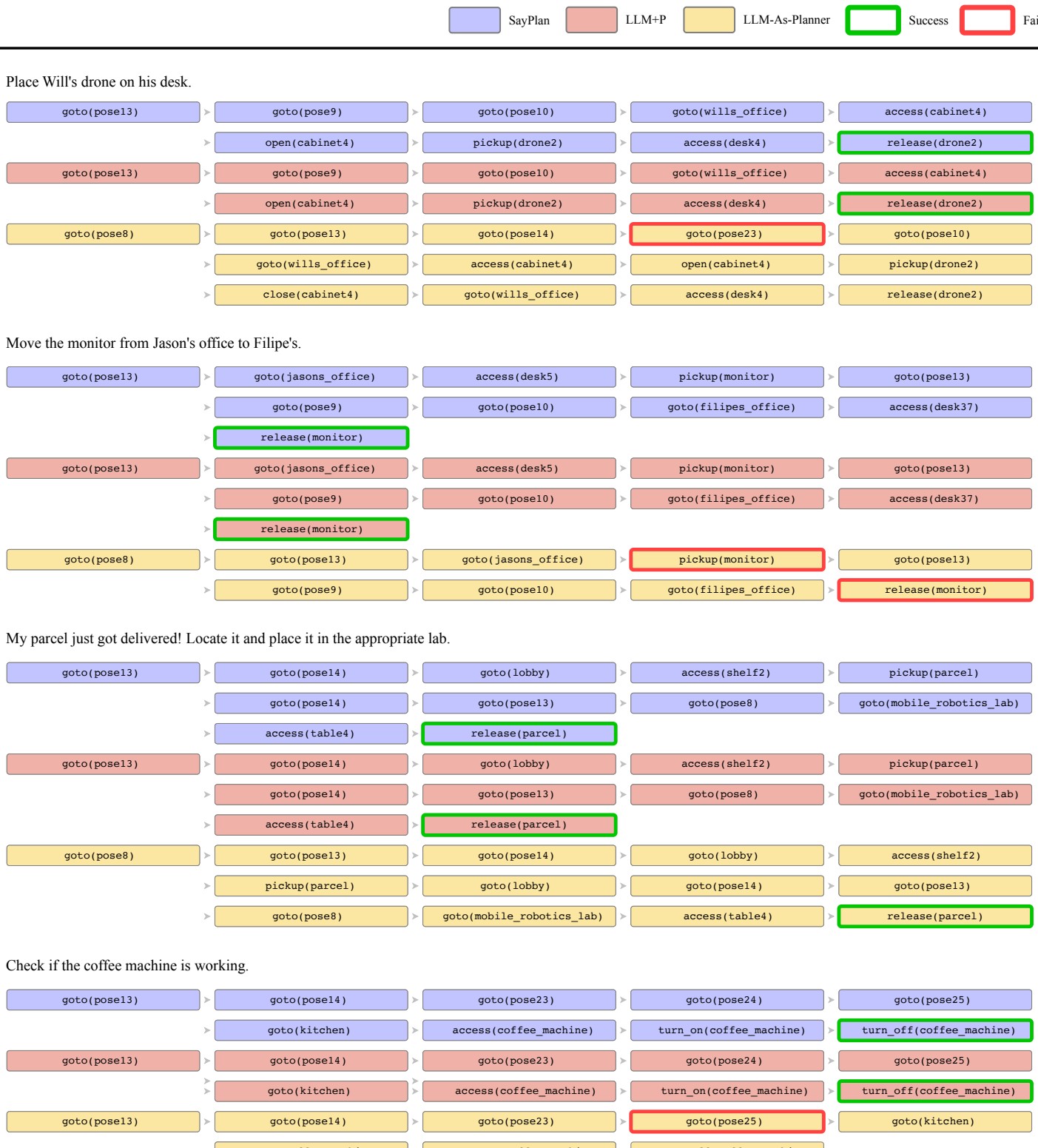

Move the monitor from Jason's office to Filipe's.

My parcel just got delivered! Locate it and place it in the appropriate lab.

Check if the coffee machine is working.

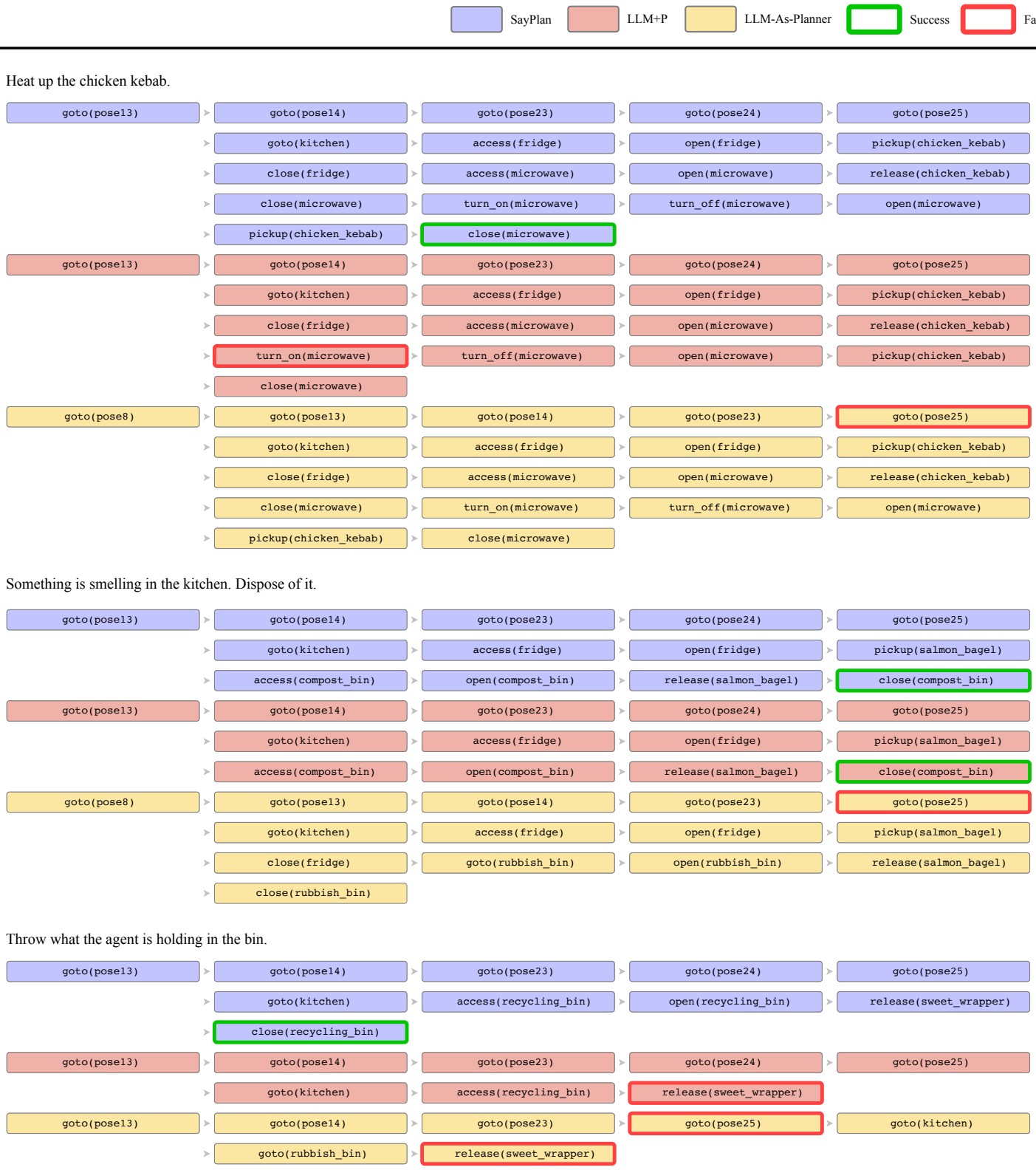

| | SayPlan | | LLM+P | | LLM-As-Planner | | Success | | Fail |
|---|---|---|---|---|---|---|---|---|---|

Heat up the noodles in the fridge, and place it somewhere where I can enjoy it.

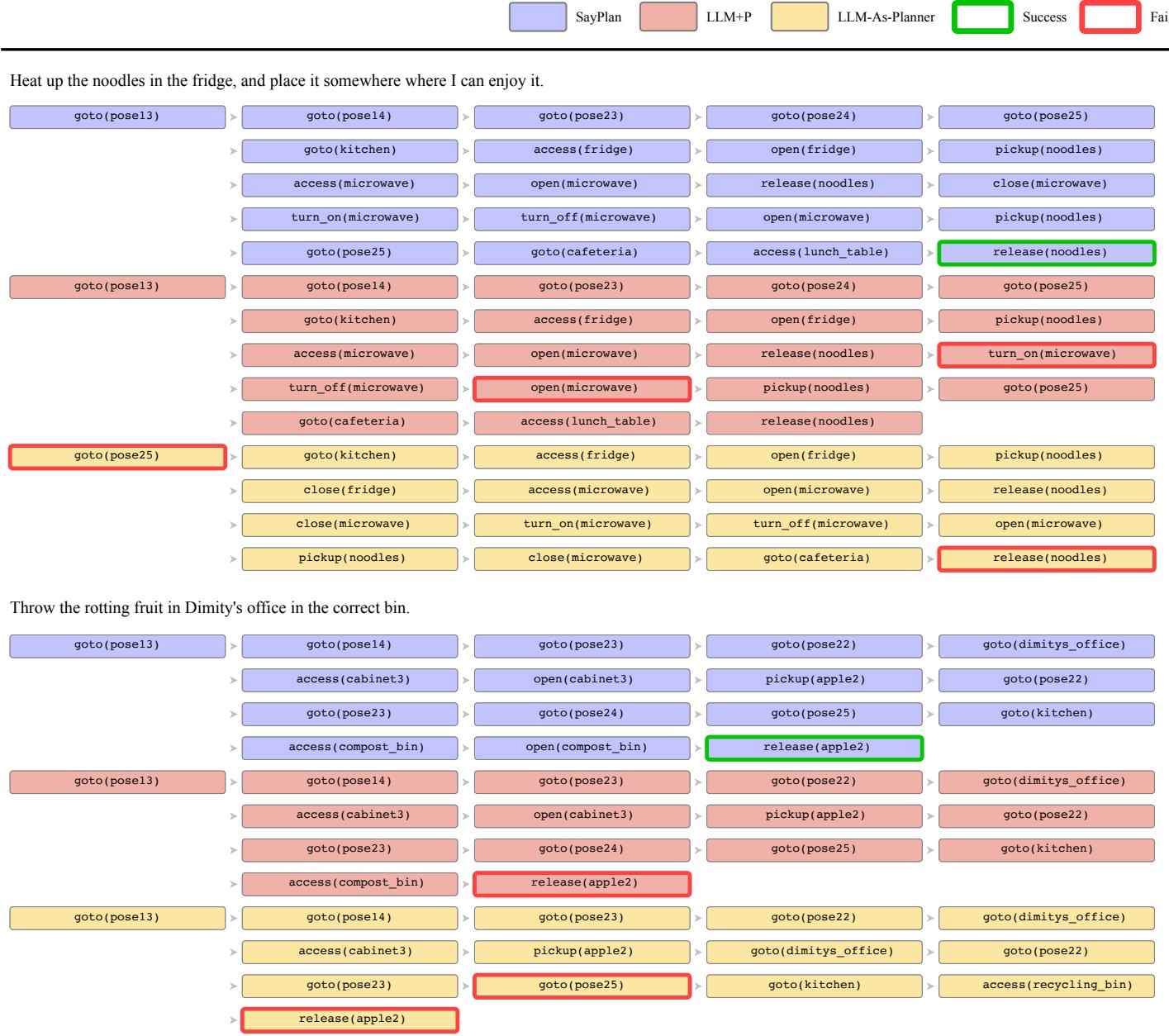

Throw the rotting fruit in Dimity's office in the correct bin.

Wash all the dishes on the lunch table. Once finished, place all the clean cutlery in the drawer.

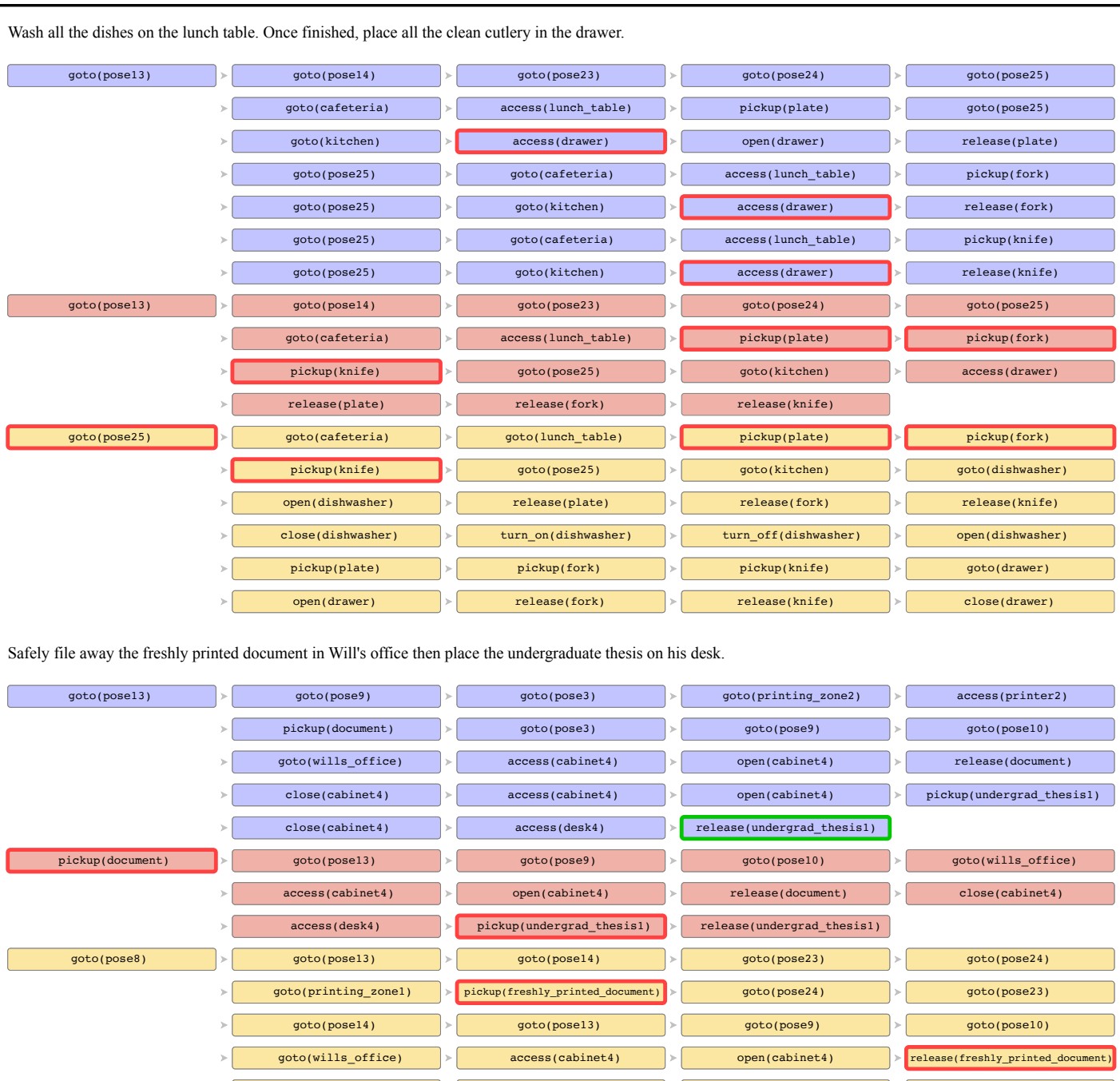

Safely file away the freshly printed document in Will's office then place the undergraduate thesis on his desk.

Make Niko a coffee and place the mug on his desk.

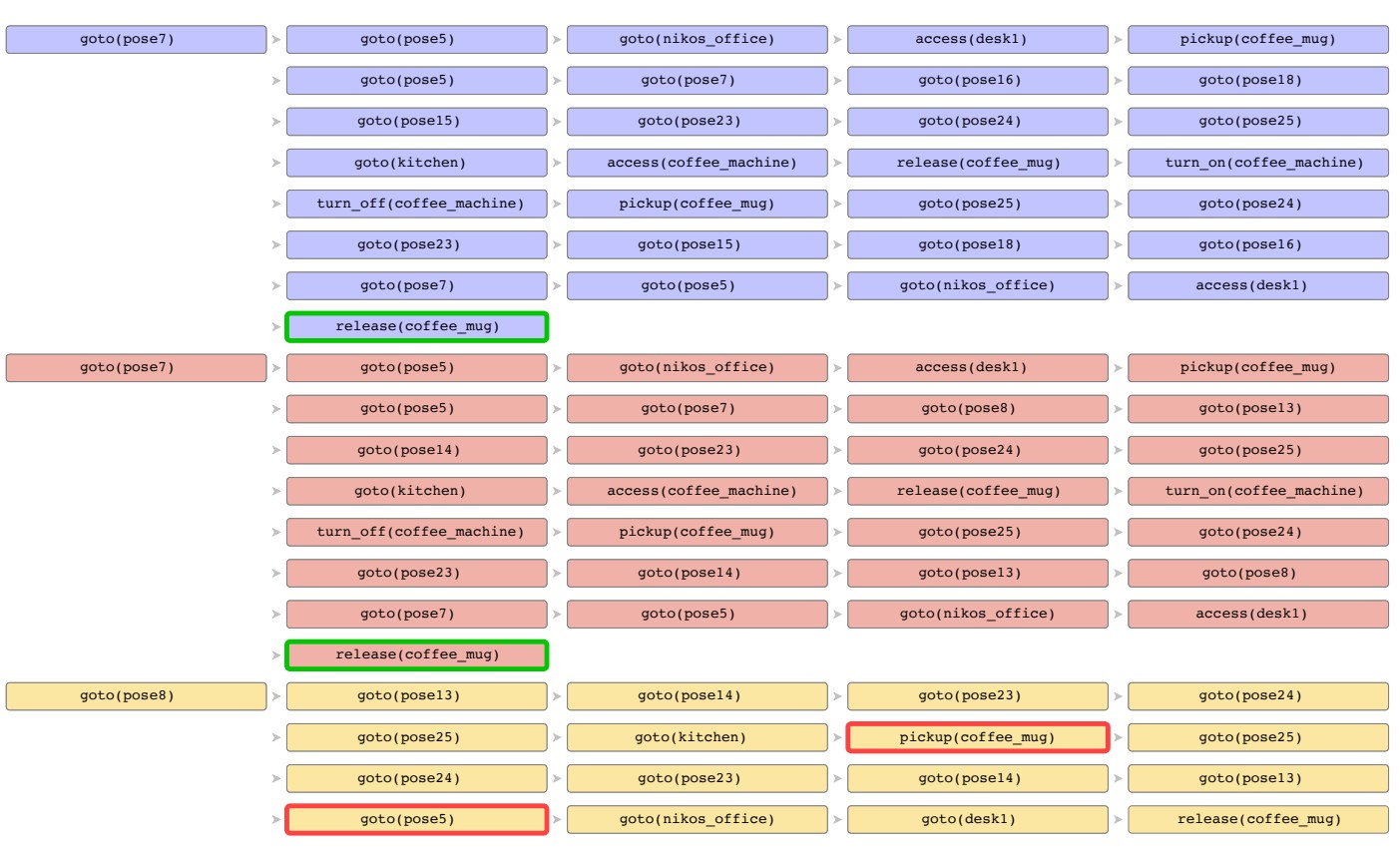

Someone has thrown items in the wrong bins. Correct this.

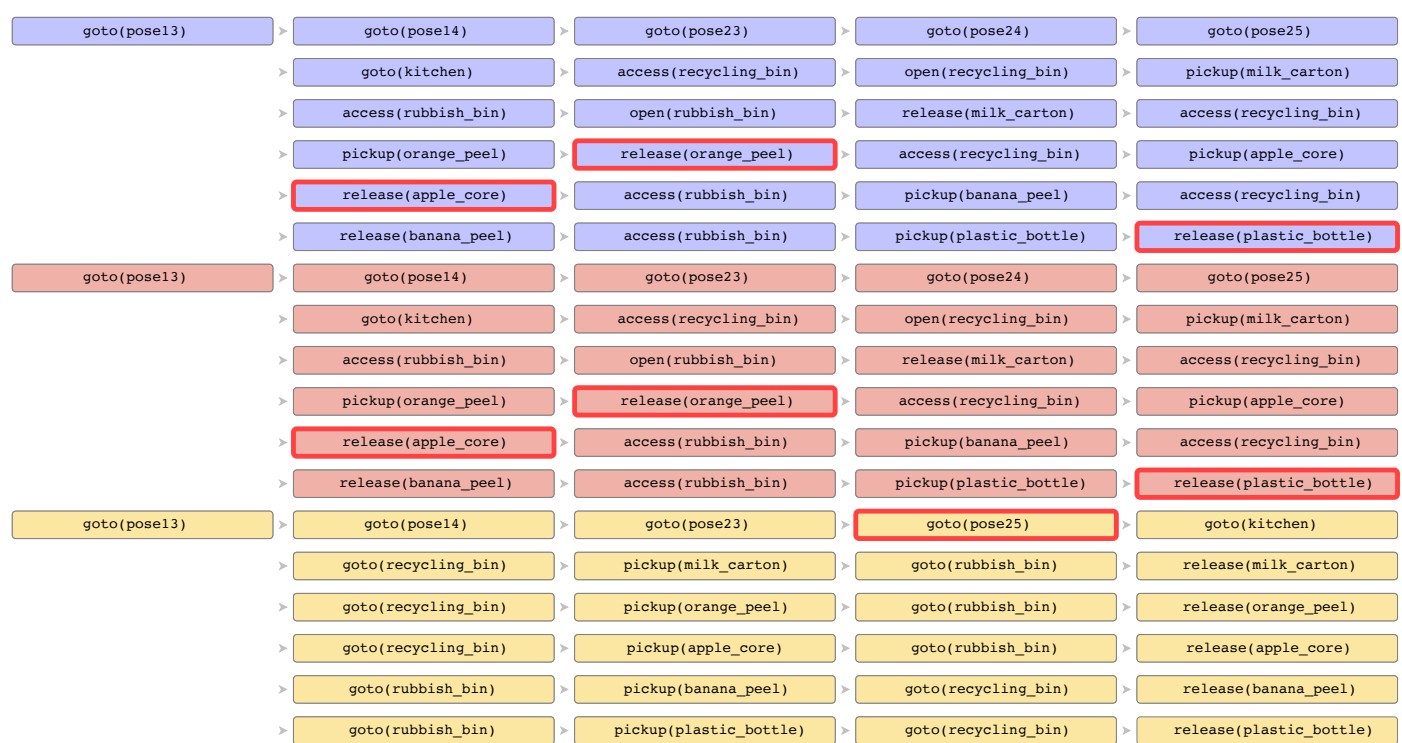

Tobi spilt soda on his desk. Throw away the can and take him something to clean with.

| | | | | |
|---|---|---|---|---|
| goto(pose7) | goto(pose5) | goto(pose2) | goto(pose1) | goto(tobis_office) |
| | access(desk38) | pickup(pepsi) | goto(pose1) | goto(pose2) |
| | goto(pose3) | goto(pose9) | goto(pose13) | goto(pose14) |
| | goto(pose23) | goto(pose24) | goto(pose25) | goto(kitchen) |
| | access(recycling_bin) | open(recycling_bin) | release(pepsi) | goto(pose25) |
| | goto(pose24) | goto(supplies_station) | access(cupboard1) | open(cupboard1) |
| | pickup(paper_towel) | goto(pose24) | goto(pose23) | goto(pose14) |
| | goto(pose13) | goto(pose9) | goto(pose3) | goto(pose2) |
| | goto(pose1) | goto(tobis_office) | access(desk38) | release(paper_towel) [Success] |

| | | | | |
|---|---|---|---|---|
| goto(pose7) | goto(pose5) | goto(pose2) | goto(pose1) | goto(tobis_office) |
| | access(desk38) | pickup(pepsi) | goto(pose1) | goto(pose2) |
| | goto(pose3) | goto(pose9) | goto(pose13) | goto(pose14) |
| | goto(pose23) | goto(pose24) | goto(pose25) | goto(kitchen) |
| | access(recycling_bin) | release(pepsi) [Fail] | goto(pose25) | goto(pose24) |
| | goto(supplies_station) | access(cupboard1) | pickup(paper_towel) | goto(pose24) |
| | goto(pose23) | goto(pose14) | goto(pose13) | goto(pose9) |
| | goto(pose3) | goto(pose2) | goto(pose1) | goto(tobis_office) |
| | release(paper_towel) [Fail] | | | |

| | | | | |
|---|---|---|---|---|
| goto(pose8) | goto(pose13) | goto(tobis_office) [Fail] | access(desk38) | pickup(pepsi) |
| | goto(tobis_office) | goto(pose1) | goto(pose2) | goto(pose5) |
| | goto(kitchen) [Fail] | access(recycling_bin) | release(pepsi) [Fail] | goto(kitchen) |
| | goto(pose5) | goto(pose2) | goto(pose24) | goto(supplies_station) |
| | access(cupboard1) | pickup(paper_towel) | goto(supplies_station) | goto(pose24) |
| | goto(pose2) | goto(pose5) | goto(pose1) | goto(tobis_office) |
| | access(desk38) | release(paper_towel) | | |

I want to make a sandwich. Place all the ingredients on the lunch table.

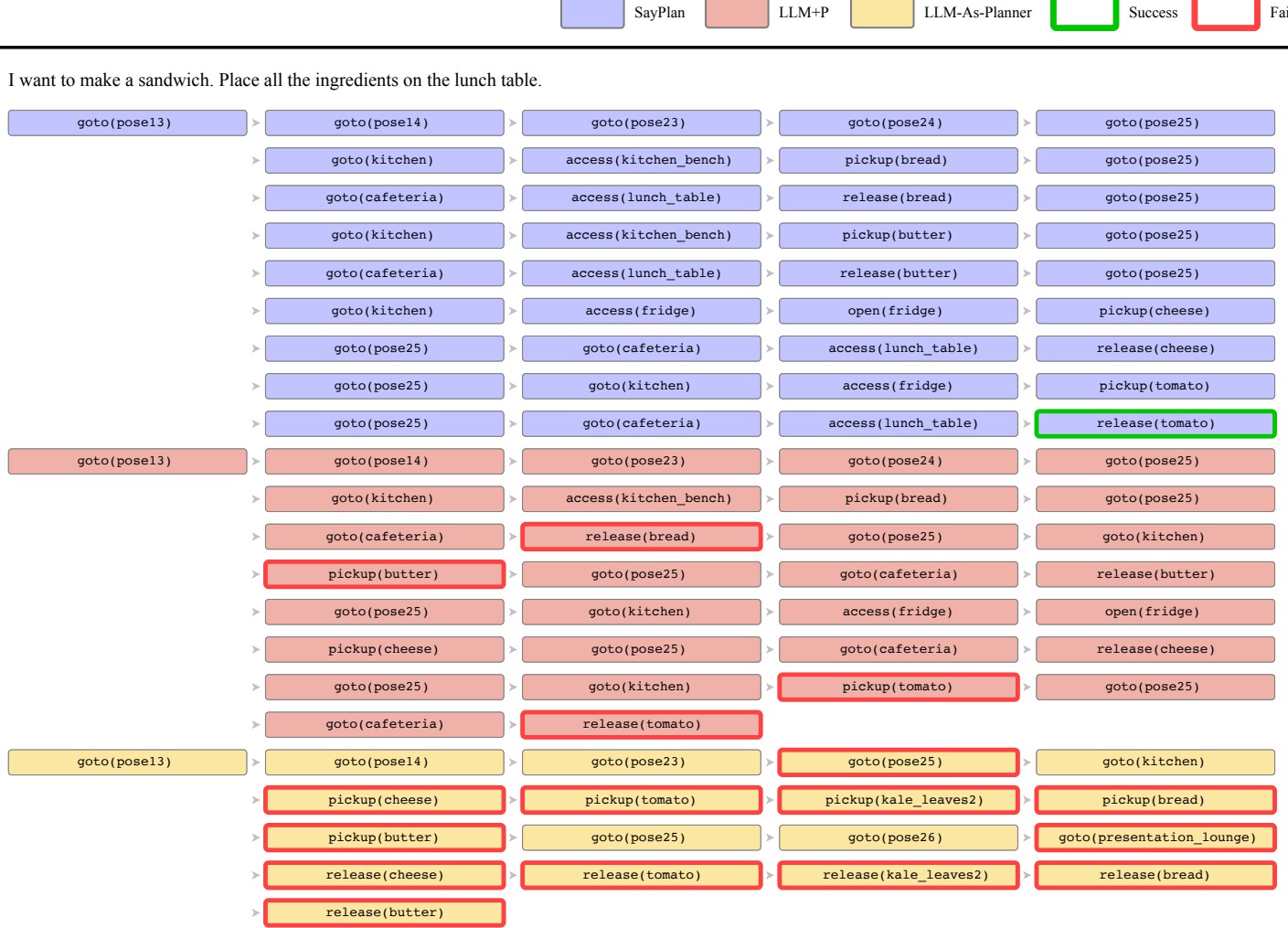

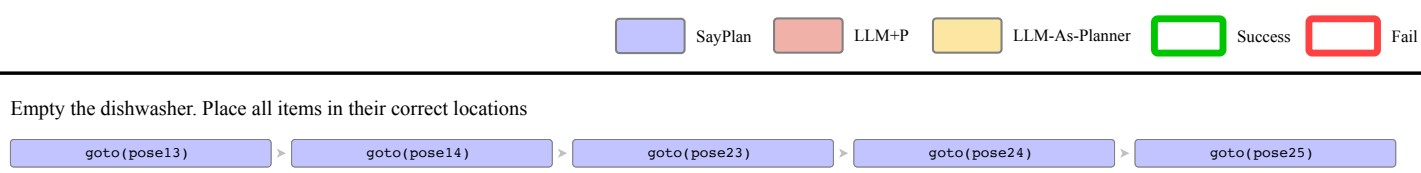

Empty the dishwasher. Place all items in their correct locations

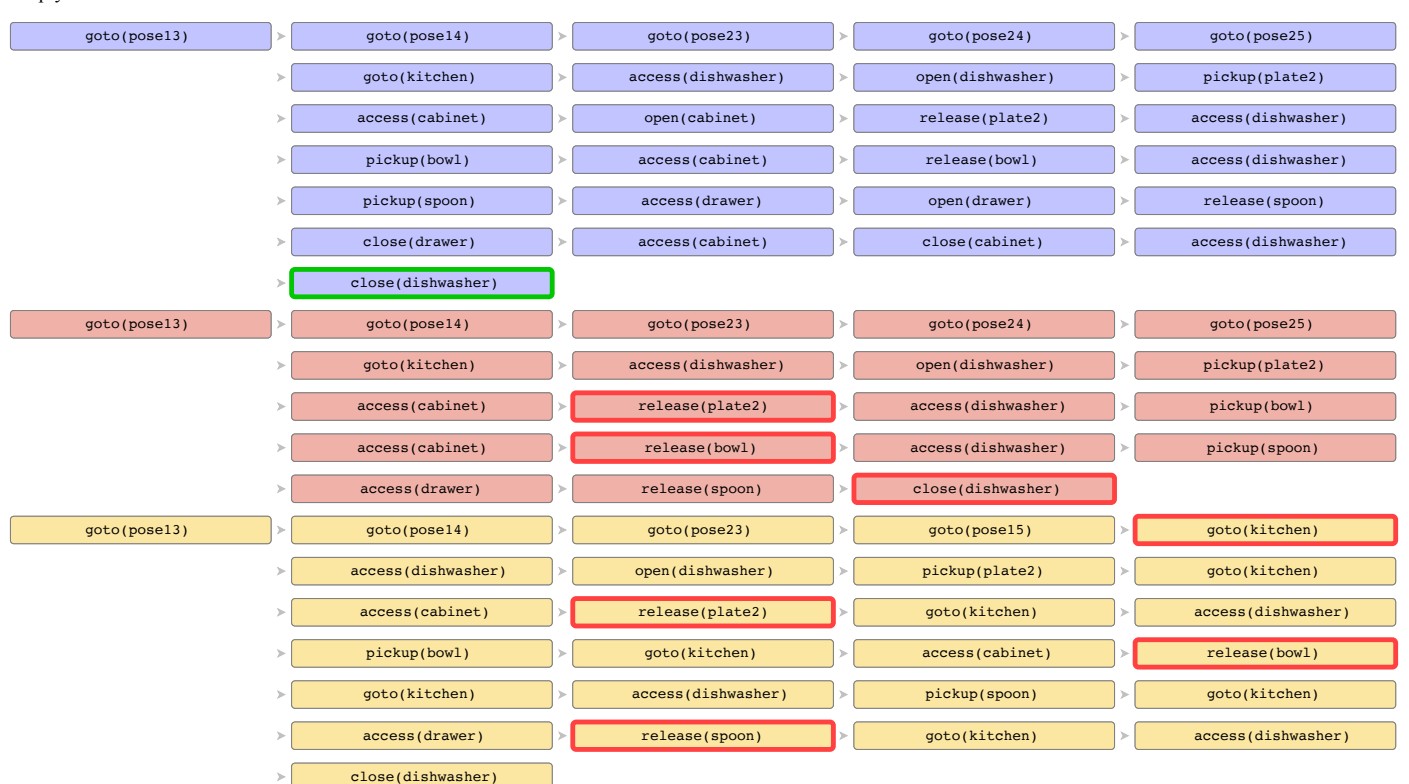

A delegation of project partners is arriving soon. We want to serve them snacks and non-alcoholic drinks. Prepare everything in the largest meeting room. Use items found in the supplies room only.

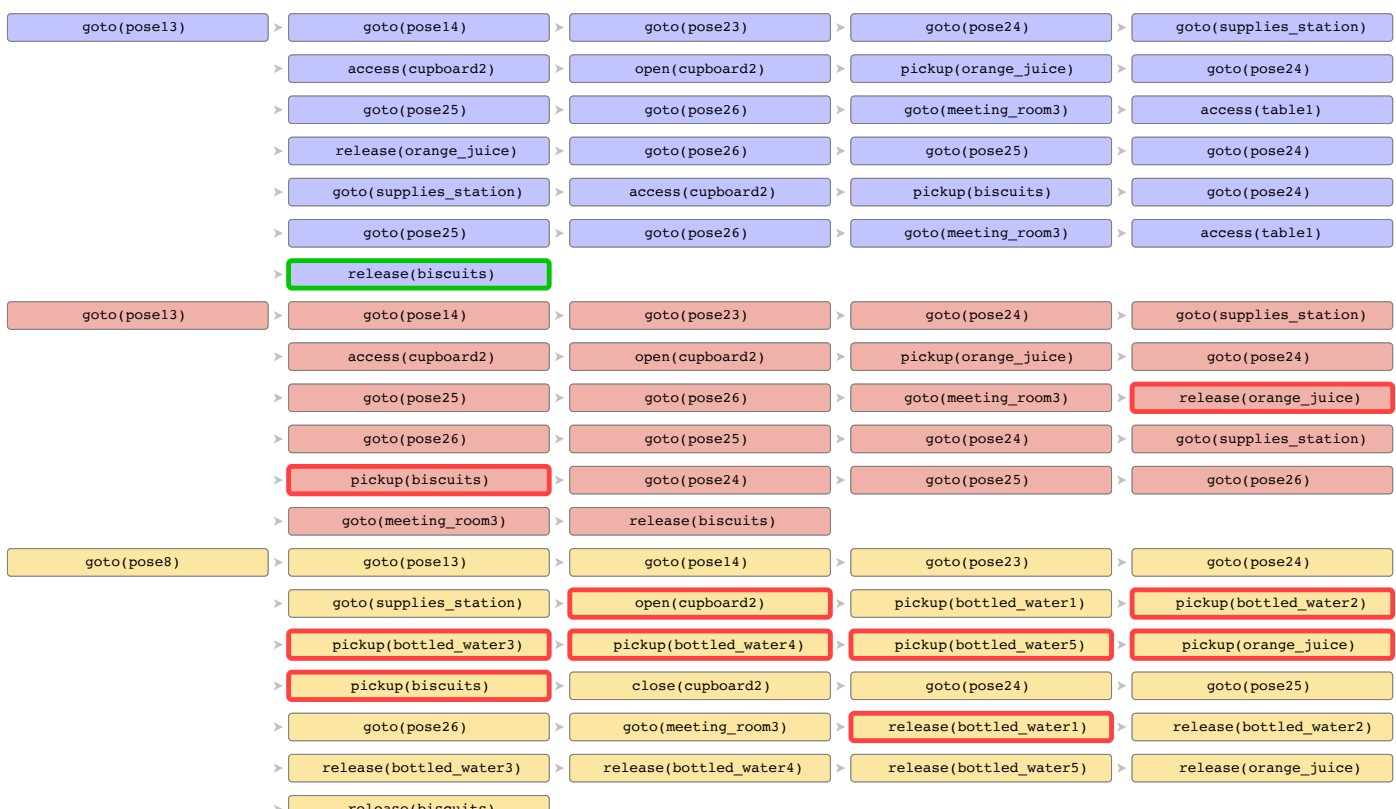

Serve bottled water to the attendees who are seated in meeting room 1. Each attendee can only receive a single bottle of water.

**SayPlan (row 1):**

goto(pose13) → goto(pose14) → goto(pose23) → goto(pose24) → goto(supplies_station)
→ access(cupboard2) → open(cupboard2) → pickup(bottled_water1) → goto(pose24)
→ goto(pose23) → goto(pose14) → goto(pose13) → goto(pose9)
→ goto(pose3) → goto(pose2) → goto(meeting_room1) → access(table5)
→ release(bottled_water1) → goto(pose2) → goto(pose3) → goto(pose9)
→ goto(pose13) → goto(pose14) → goto(pose23) → goto(pose24)
→ goto(supplies_station) → access(cupboard2) → pickup(bottled_water2) → goto(pose24)
→ goto(pose23) → goto(pose14) → goto(pose13) → goto(pose9)
→ goto(pose3) → goto(pose2) → goto(meeting_room1) → access(table5)
→ release(bottled_water2) → goto(pose2) → goto(pose3) → goto(pose9)
→ goto(pose13) → goto(pose14) → goto(pose23) → goto(pose24)
→ goto(supplies_station) → access(cupboard2) → pickup(bottled_water3) → goto(pose24)
→ goto(pose23) → goto(pose14) → goto(pose13) → goto(pose9)
→ goto(pose3) → goto(pose2) → goto(meeting_room1) → access(table5)
→ release(bottled_water3) **[Success]**

**LLM+P (row 2):**

goto(pose13) → goto(pose14) → goto(pose23) → goto(pose24) → goto(supplies_station)
→ access(cupboard2) → pickup(bottled_water1) **[Fail]** → goto(pose24) → goto(pose23)
→ goto(pose14) → goto(pose13) → goto(pose9) → goto(pose3)
→ goto(pose2) → goto(meeting_room1) → release(bottled_water1) **[Fail]** → goto(pose2)
→ goto(pose3) → goto(pose9) → goto(pose13) → goto(pose14)
→ goto(pose23) → goto(pose24) → goto(supplies_station) → access(cupboard2)
→ pickup(bottled_water2) **[Fail]** → goto(pose24) → goto(pose23) → goto(pose14)
→ goto(pose13) → goto(pose9) → goto(pose3) → goto(pose2)
→ goto(meeting_room1) → release(bottled_water2) **[Fail]** → goto(pose2) → goto(pose3)
→ goto(pose9) → goto(pose13) → goto(pose14) → goto(pose23)
→ goto(pose24) → goto(supplies_station) → access(cupboard2) → pickup(bottled_water3) **[Fail]**
→ goto(pose24) → goto(pose23) → goto(pose14) → goto(pose13)
→ goto(pose9) → goto(pose3) → goto(pose2) → goto(meeting_room1)
→ release(bottled_water3) **[Fail]**

**LLM-As-Planner (row 3):**

goto(pose8) → goto(pose13) → goto(pose14) → goto(pose23) → goto(supplies_station)
→ pickup(bottled_water1) **[Fail]** → goto(pose23) → goto(pose14) → goto(pose1)
→ goto(meeting_room1) → release(bottled_water1) **[Fail]** → goto(pose1) → goto(pose14)
→ goto(pose23) → goto(supplies_station) → pickup(bottled_water2) **[Fail]** → goto(pose23)
→ goto(pose14) → goto(pose1) → goto(meeting_room1) → release(bottled_water2) **[Fail]**
→ goto(pose1) → goto(pose14) → goto(pose23) → goto(supplies_station)
→ pickup(bottled_water3) **[Fail]** → goto(pose23) → goto(pose14) → goto(pose1)
→ goto(meeting_room1) → release(bottled_water3) **[Fail]**

Locate all 6 complimentary t-shirts given to the PhD students and place them on the shelf in admin.

**SayPlan (row 1):**

| | | | | |
|---|---|---|---|---|
| goto(pose7) | goto(pose5) | goto(pose6) | goto(phd_bay1) | access(desk9) |
| | pickup(complimentary_tshirt1) | goto(pose6) | goto(pose5) | goto(pose7) |
| | goto(pose16) | goto(pose18) | goto(admin) | access(shelf) |
| | release(complimentary_tshirt1) | goto(pose18) | goto(pose16) | goto(pose7) |
| | goto(pose5) | goto(pose6) | goto(phd_bay1) | access(desk10) |
| | pickup(complimentary_tshirt2) | goto(pose6) | goto(pose5) | goto(pose7) |
| | goto(pose16) | goto(pose18) | goto(admin) | access(shelf) |
| | release(complimentary_tshirt2) | goto(pose18) | goto(pose16) | goto(pose17) |
| | goto(phd_bay2) | access(desk15) | pickup(complimentary_tshirt3) | goto(pose17) |
| | goto(pose16) | goto(pose18) | goto(admin) | access(shelf) |
| | release(complimentary_tshirt3) | goto(pose18) | goto(pose16) | goto(pose17) |
| | goto(phd_bay2) | access(desk18) | pickup(complimentary_tshirt4) | goto(pose17) |
| | goto(pose16) | goto(pose18) | goto(admin) | access(shelf) |
| | release(complimentary_tshirt4) | goto(pose18) | goto(pose16) | goto(pose17) |
| | goto(phd_bay2) | access(desk18) | pickup(complimentary_tshirt6) | goto(pose17) |
| | goto(pose16) | goto(pose18) | goto(admin) | access(shelf) |
| | release(complimentary_tshirt6) | goto(pose18) | goto(pose19) | goto(phd_bay4) |
| | access(desk25) | pickup(complimentary_tshirt5) | goto(pose19) | goto(pose18) |
| | goto(admin) | access(shelf) | release(complimentary_tshirt5) [Success] | |

**LLM+P (row 2):**

| | | | | |
|---|---|---|---|---|
| goto(pose7) | goto(pose5) | goto(pose6) | goto(phd_bay1) | access(desk9) |
| | pickup(complimentary_tshirt1) | goto(pose6) | goto(pose5) | goto(pose7) |
| | goto(pose16) | goto(pose18) | goto(admin) | access(shelf) |
| | release(complimentary_tshirt1) | goto(pose18) | goto(pose16) | goto(pose7) |
| | goto(pose5) | goto(pose6) | goto(phd_bay1) | access(desk10) |
| | pickup(complimentary_tshirt2) | goto(pose6) | goto(pose5) | goto(pose7) |
| | goto(pose16) | goto(pose18) | goto(admin) | access(shelf) |
| | release(complimentary_tshirt2) | goto(pose18) | goto(pose16) | goto(pose17) |
| | goto(phd_bay2) | access(desk15) | pickup(complimentary_tshirt3) | goto(pose17) |
| | goto(pose16) | goto(pose18) | goto(admin) | access(shelf) |
| | release(complimentary_tshirt3) | goto(pose18) | goto(pose16) | goto(pose17) |
| | goto(phd_bay2) | access(desk18) | pickup(complimentary_tshirt4) | goto(pose17) |
| | goto(pose16) | goto(pose18) | goto(admin) | access(shelf) |
| | release(complimentary_tshirt4) | access(desk18) [Fail] | pickup(complimentary_tshirt6) | access(shelf) |
| | release(complimentary_tshirt6) | goto(pose18) | goto(pose19) | goto(phd_bay4) |
| | access(desk25) | pickup(complimentary_tshirt5) | goto(pose19) | goto(pose18) |
| | goto(admin) | access(shelf) | release(complimentary_tshirt5) | |

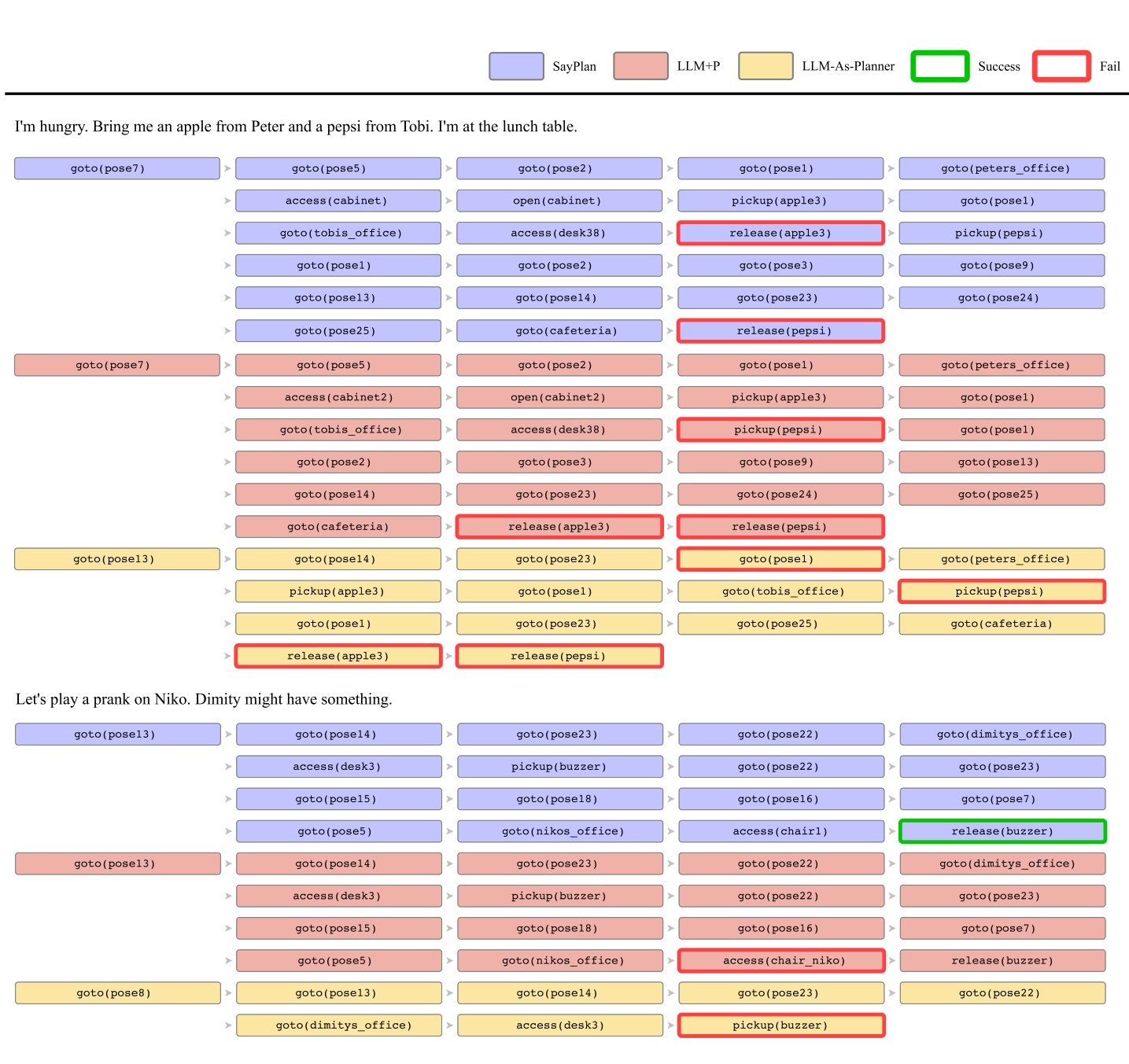

I'm hungry. Bring me an apple from Peter and a pepsi from Tobi. I'm at the lunch table.

Let's play a prank on Niko. Dimity might have something.

Table 19: **Causal Planning Evaluation.** Task planning action sequences generated for a mobile manipulator robot to follow for both the simple and long-horizon planning instruction sets.

# H  Scalability Ablation Study

In this study, we evaluate the ability of SayPlan and the underlying LLM to reason over larger-scale scene graphs. More specifically, as SayPlan's initial input is a collapsed 3DSG, we explore how increasing the number of nodes in this base environment impacts the ability of the LLM to attend to the relevant parts of the scene graph for both semantic search and iterative replanning.

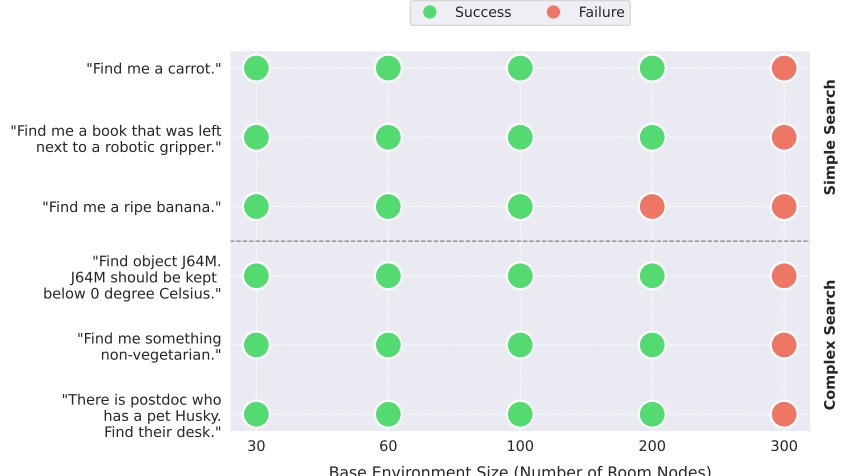

Figure 7: **Evaluating the performance of the underlying LLMs semantic search capabilities as the scale of the environment increases.** For the office environment used in this study, we are primarily interested in the number of room nodes present in the collapsed form of the 3DSG.

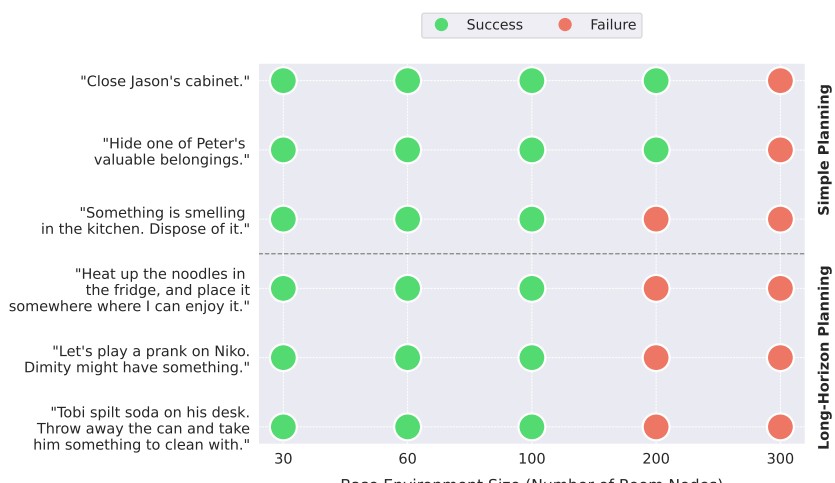

Figure 8: **Evaluating the performance of SayPlan's causal planning capabilities as the scale of the environment increases.** For the office environment used in this study, we are primarily interested in the number of room nodes present in the collapsed form of the 3DSG.

We note here that all the failures that occurred across both semantic search and iterative replanning were a result of the LLM's input exceeding the maximum token limits – in the case of `GPT-4` this corresponded to 8192 tokens. With regard to the scalability to larger environments, this is an important observation as it indicates that the LLM's reasoning capabilities or ability to attend to the relevant parts of the 3DSG is not significantly impacted by the presence of "noisy" or increasing number of nodes. One potential downside to larger environments however is the increased number of steps required before semantic search converges. As more semantically relevant floor or room nodes enter the scene, each one of these may be considered by the LLM for exploration.

# I    Real World Execution of a Generated Long Horizon Plan.

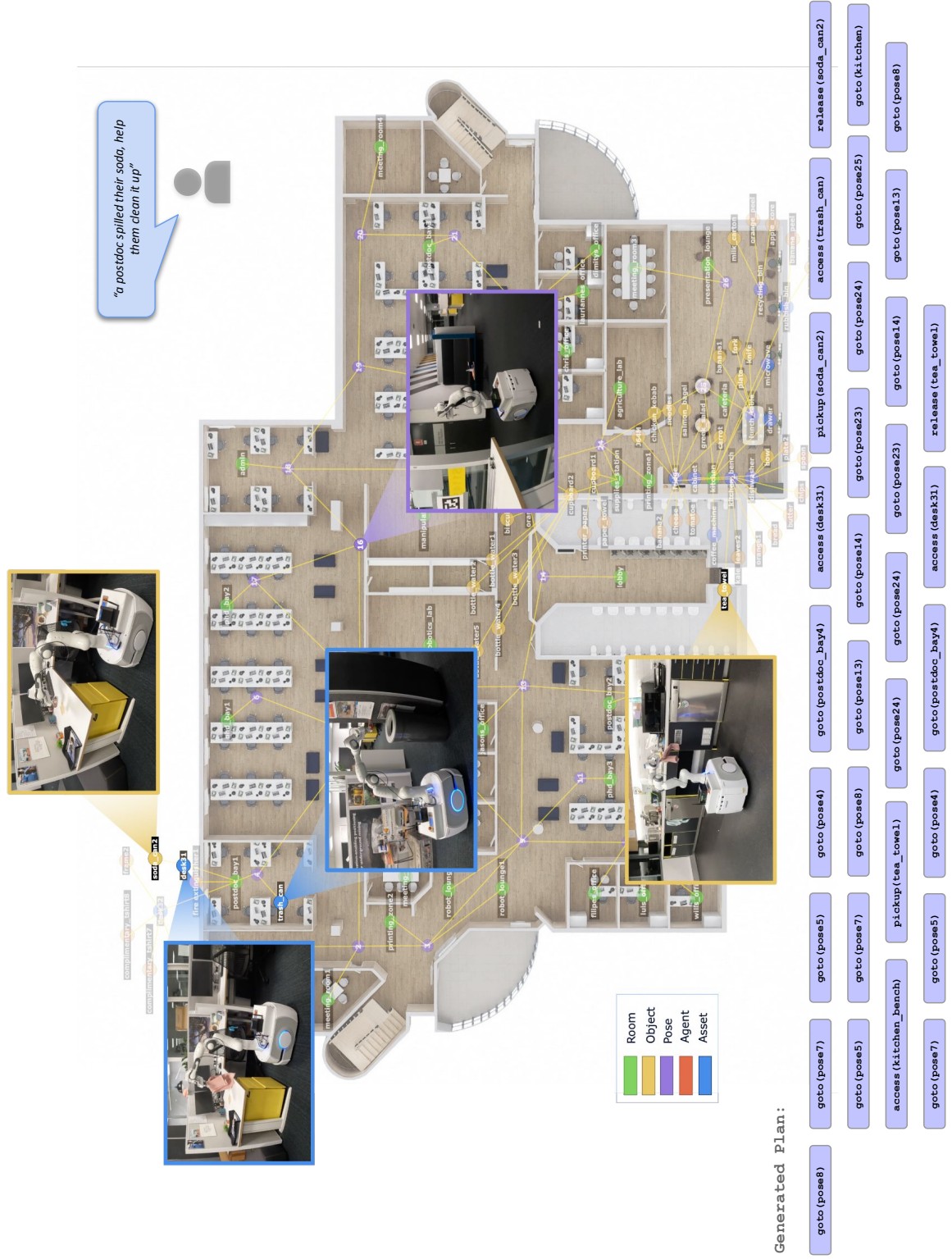

Figure 9: **Real World Execution of a Generated Long Horizon Plan.** Execution of a generated and validated task plan on a real-world mobile manipulator robot.

## J Input Prompt Structure

Input prompt passed to the LLM for SayPlan. Note that the components highlighted in violet represent static components of the prompt that remain fixed throughout both the semantic search and iterative replanning phases of SayPlan.

---

**Agent Role:** You are an excellent graph planning agent. Given a graph representation of an environment, you can explore the graph by expanding nodes to find the items of interest. You can then use this graph to generate a step-by-step task plan that the agent can follow to solve a given instruction.

**Environment Functions:**
*goto(<pose>):* Move the agent to any room node or pose node.
*access(<asset>):* Provide access to the set of affordances associated with an asset node and its connected objects.
*pickup(<object>):* Pick up an accessible object from the accessed node.
*release(<object>):* Release grasped object at an asset node.
*turn_on/off(<object>):* Toggle object at agent's node, if accessible and has affordance.
*open/close(<asset>):* Open/close asset at agent's node, affecting object accessibility.
*done():* Call when the task is completed.

**Environment State:**
*ontop_of(<asset>):* Object is located on <asset>
*inside_of(<asset>):* Object is located inside <asset>
*inside_hand:* Object is currently being grasped by the robot/agent
*closed:* Asset can be opened
*open:* Asset can be closed or kept open
*on:* Asset is currently on
*off:* Asset is currently off
*accessible:*  The object is not accessible if it is inside an asset and the asset state is "closed".

**Environment API:**
*expand_node(<node>):* Reveal assets/objects connected to a room/floor node.
*contract_node(<node>):* Hide assets/objects, reducing graph size for memory constraints.
*verify_plan():* Verify generated plan in the scene graph environment.

**Output Response Format:**
{*chain_of_thought:* break your problem down into a series of intermediate reasoning steps to help you determine your next command,
*reasoning:* justify why the next action is important
*mode:* "exploring" OR "planning"
*command:* {``command_name": Environment API call
            "node_name": node to perform an operation on
            "plan": task plan if in planning mode}}
**Example:** <see Appendix K and L>
**Instruction:** Natural language description of the task
**3D Scene Graph:** Text-serialised JSON description of a 3D scene graph
**Memory:** History of previously expanded nodes
**Feedback:** External textual feedback from scene graph simulator
Ensure the response can be parsed by Python json.loads.

---

## K Example Input-Output Interaction Between LLM and Scene Graph Simulator During Semantic Search

**Instruction**: make a coffee for Tom and place it in his room
**3D Scene Graph**: {nodes: {room: [{id: bobs_room}, {id: toms_room}, {id: jacks_room}, {id: kitchen}, {id: livingroom}], pose: [{id: pose1}, {id: pose2}, {id: pose3}, {id: pose4}, {id: pose5}], agent: [{location: bobs_room, id: agent}]}, links: [bobs_room↔pose1, bobs_room↔agent, toms_room↔pose1, toms_room↔pose2, toms_room↔pose5, jacks_room↔pose2, jacks_room↔pose3, kitchen↔pose3, kitchen↔pose4, kitchen↔pose5, livingroom↔pose4]}
**Memory**: []
**SayPlan [Mode->Search]:** chain-of-thought: explore graph for coffee mug, coffee machine and somewhere to place the coffee -> perform common sense exploration first -> explore toms room -> explore kitchen -> if items not found then explore other nodes
reasoning: i will expand toms room
command: expand(toms_room)
**Scene Graph Simulator:** <Updated 3D scene graph>
**Memory**: [toms_room]
**SayPlan [Mode->Search]:** chain-of-thought: i have found a wardrobe in tom's room -> leave this node expanded -> the coffee mug is not in his room -> still have not found the coffee machine -> kitchen might have coffee machine and coffee mug -> explore this node next
reasoning: i will expand the kitchen next
command: expand(kitchen)
**Scene Graph Simulator:** <Updated 3D scene graph>
**Memory**: [toms_room, kitchen]
**SayPlan [Mode->Search]:** chain-of-thought: i've found the coffee machine in the kitchen -> the coffee mug however is not in the kitchen -> i will explore other unexpanded nodes now for the coffee mug -> jacks room, bobs room and the living room are unexplored yet -> i will try jack"s room next
reasoning: i will expand jacks room next
command: expand(jacks_room)
**Scene Graph Simulator:** <Updated 3D scene graph>
**Memory**: [toms_room, kitchen, jacks_room]
**SayPlan [Mode->Search]:** chain-of-thought: jacks room does not have anything relevant towards solving this task -> I should contract this node to save token memory -> i will explore other unexplored nodes after doing this
reasoning: contract jack's room to save memory
command: contract(jacks_room)
**Scene Graph Simulator:** <Updated 3D scene graph>
**Memory**: [toms_room, kitchen, jacks_room]
**SayPlan [Mode->Search]:** chain-of-thought: i have found a wardrobe in tom's room and coffee machine in the kitchen -> the coffee mug is still required for the task -> explore a new room next
reasoning: i will expand bob's room as it is unexplored
command: expand(bobs_room)
**Scene Graph Simulator:** <Updated 3D scene graph>
**Memory**: [toms_room, kitchen, jacks_room, bobs_room]
**SayPlan [Mode->Search]:** chain-of-thought: i have found the coffee mug, the coffee machine and tom's wardrobe on the graph -> search complete -> I will switch to planning mode-> collect coffee mug -> generate plan for making coffee -> place coffee mug on Tom's wardrobe
reasoning: I will generate a task plan using the identified subgraph

## L  Example Input-Output Interaction Between LLM and Scene Graph Environment During Iterative Replanning

```
Instruction: make a coffee for Tom and place it in his room
3D Scene Graph: {nodes: {room: [{id: bobs_room}, {id: toms_room}, {id:
jacks_room}, {id: kitchen}, {id: livingroom}], pose: [{id: pose1},
{id: pose2}, {id: pose3}, {id: pose4}, {id: pose5}], agent: [{location:
bobs_room, id: agent}], asset: [{room: toms_room, state: free,
affordances: [release], id: bed2}, {room: toms_room, state: closed,
affordances: [open, close, release], id: wardrobe2}, {room: kitchen,
state: closed, affordances: [open, close, release], id: fridge},
{room: kitchen, affordances: [turn_on, turn_off], state: off, id:
coffee_machine}, {room: bobs_room, state: free, affordances: [release],
id: bed1}, {room: bobs_room, state: closed, affordances: [open, close,
release], id: wardrobe1}], object: [{affordances: [pickup], state:
inside_of(wardrobe1), attributes: "blue", id: coffee_mug}]}, links:
[bobs_room↔pose1, bobs_room↔agent, bobs_room↔bed1, bobs_room↔wardrobe1,
toms_room↔pose1, toms_room↔pose2, toms_room↔pose5, toms_room↔bed2,
toms_room↔wardrobe2, jacks_room↔pose2, jacks_room↔pose3, kitchen↔pose3,
kitchen↔pose4, kitchen↔pose5, kitchen↔fridge, kitchen↔coffee_machine,
livingroom↔pose4, wardrobe1↔coffee_mug]}
Memory: [toms_room, kitchen, jacks_room, bobs_room]
SayPlan [Mode->Planning]: chain-of-thought: "i have found the coffee mug,
the coffee machine and tom's wardrobe on the graph -> search complete -> I
will switch to planning mode-> collect coffee mug -> generate plan for making
coffee -> place coffee mug on Tom's wardrobe"
reasoning: I will generate a task plan using the identified subgraph
plan: [goto(bobs_room) > access(wardrobe1) > pickup(coffee_mug)
> goto(kitchen) > access(coffee_machine) > release(coffee_mug) >
turn_on(coffee_machine) > turn_off(coffee_machine) > pickup(coffee_mug) >
goto(toms_room) > access(wardrobe2) > release(coffee_mug) > done]
Path Planner:plan: [goto(pose3) > goto(pose4) > goto(pose1) >
goto(bobs_room) > access(wardrobe1) > pickup(coffee_mug) > goto(pose2) >
goto(pose5) > goto(kitchen) > access(coffee_machine) > release(coffee_mug)
> turn_on(coffee_machine) > turn_off(coffee_machine) > pickup(coffee_mug)
> goto(pose5) > goto(pose2) > goto(toms_room) > access(wardrobe2) >
release(coffee_mug) > done]
Scene Graph Simulator (Feedback): "coffee mug is not accessible"
Memory: [toms_room, kitchen, jacks_room, bobs_room]
SayPlan [Mode->Planning]: chain-of-thought: "the coffee mug was not
accessible -> the wardrobe it is in is currently closed -> I update the plan
to open the wardrobe first -> generate plan for making coffee -> place coffee
mug on Tom's wardrobe"
reasoning: I will generate a task plan using the identified subgraph and the
feedback from the simulator
plan: [goto(bobs_room) > access(wardrobe1) > open(wardrobe1) >
pickup(coffee_mug) > goto(kitchen) > access(coffee_machine) >
release(coffee_mug) > turn_on(coffee_machine) > turn_off(coffee_machine)
> pickup(coffee_mug) > goto(toms_room) > access(wardrobe2) >
release(coffee_mug) > done]
Path Planner:plan: [goto(pose3) > goto(pose4) > goto(pose1) >
goto(bobs_room) > access(wardrobe1) > open(wardrobe1) > pickup(coffee_mug)
> goto(pose2) > goto(pose5) > goto(kitchen) > access(coffee_machine) >
release(coffee_mug) > turn_on(coffee_machine) > turn_off(coffee_machine)
> pickup(coffee_mug) > goto(pose5) > goto(pose2) > goto(toms_room) >
access(wardrobe2) > release(coffee_mug) > done]
Scene Graph Simulator: Plan Verified
```

