# OpenReview forum: "SayPlan: Grounding Large Language Models using 3D Scene Graphs for Scalable Robot Task Planning"
_robot-learning.org/CoRL/2023/Conference — CoRL 2023 Oral_

### Official Review · Reviewer_QfzX · 2023-07-06

**Confidence:** 4
**Originality:** Excellent
**Technical Quality:** Very Good
**Clarity Of Presentation:** Good
**Impact:** 4

**Recommendation:**

Strong Accept: I recommend accepting the paper and will argue for my recommendation even if other reviewers hold a different opinion.

**Review:**

Strengths – The main strength of this paper is the pre-encoding idea of the large world model leveraging the 3D scene graph and its sub-graph identification.
Weaknesses – This method cannot be scalable with more geometric or semantic attributes on each graph node or edge.

This paper needs to clarify and improve a few things.
-	The major concern is the limitation of the sub-graph selection for LLM. The 3D scene graph can hold more information this paper considers. Node and edge features may include more spatial and/or semantic information depending on the capabilities of the detectors a robot has. Due to that, the discovered subgraph may still have too much information that the LLM cannot handle. Thus, it is necessary to clarify how many nodes and features of a graph typical LLM handles.
-	Another issue is the lack of explanations. I understand the limited space makes it hard to include all the contents. However, it is really hard to understand how the semantic search work with LLM, though Appendix 1 helps. It will be great if the authors explain it with examples in the paper.
-	Algorithm 1 must be improved.


**Quality Of The Limitations Section:**

Limitations are addressed clearly

**Questions For Rebuttal:**

-	How many nodes and features of a graph typical LLM can handle? How can you handle a given with too many features?
-	What is the scene graph simulator?
-	Algorithm 1. 4-6: what is `node_id’?


**Robotics Focus:**

Sufficient demonstration on hardware

**Summary Of Paper:**

This paper aims to solve a problem of large language model (LLM) based task planning in a large-scale environment including multiple rooms and floors. The problem is hard in that the pre-encoding of all the necessary information within the LLMs may not be feasible. This paper proposes a scalable LLM-based large-scale task planning method with a 3D scene graph. The main contributions of this paper are as follows:
-	a semantic search of task-relevant subgraphs to encode the world model information to LLM
-	an iterative replanning to find feasible action plans


**Summary Of Recommendation:**

The proposed methodology is very interesting and provides the direction of using LLM with a large world model. I would like to recommend accepting this paper.

---

### Official Review · Reviewer_xLeS · 2023-07-17

**Confidence:** 4
**Originality:** Very Good
**Technical Quality:** Good
**Clarity Of Presentation:** Very Good
**Impact:** 3

**Recommendation:**

Strong Accept: I recommend accepting the paper and will argue for my recommendation even if other reviewers hold a different opinion.

**Review:**

Strengths
* This paper provides one of the first efforts to integrate 3D scene graphs with LLMs. 3DSGs are specified via a JSON-serialized format and given to the LLM. Leveraging LLMs to perform semantic search (finding task-relevant subgraphs to do search) appears novel, although integrating LLM-planning with low-level planners and feedback have been explored in prior works (e.g. InnerMonologue, as the authors cited).
* The related works on LLM planners seem sufficient.
* The methodology is technically sound overall. The experiments on two large environments with different instruction sets provide convincing evidence for the claims. The experiments especially highlight the importance of the iterative replanning process.
* Analysis of token requirements shows approach scales to large environments.
* Detailed analysis and breakdown of plan failure modes.
* The paper is well-written and easy to follow. The approach is clearly explained through diagrams and examples.

Weaknesses/Feedback
1) More analysis could be provided on how the plan quality changes with increasing environment size. It might be good to see at which scale SayPlan starts to outperform the baselines, and at which scale SayPlan performance becomes inadequate. Since a big part of the paper is about proposing a planning method that scales, providing scaling experiments would make the paper a lot stronger.
2) The only type of failure for SayPlan is hallucinating nodes that do not exist. The authors mention that additional iterative replanning iterations may improve this issue. I think providing ablations on plan performance vs. the number of iterations is a really important comparison.
3) It will be interesting to see at how many iterations do the other types of failure modes start to disappear (maybe 1 is enough? Or 2? Or is it a smoother interpolation?), and to see if it’s possible to achieve 0% hallucinating nodes with SayPlan (maybe at 20 iterations? Maybe more?). This would be a super interesting experiment that would really allow readers to understand the extent of LLMs’ self-correcting capabilities.
4) There are no comparisons to task planners that do not use LLMs. Since the scene graphs are given, are there really no other task-planning methods that can serve as a reasonable baseline?
5) Currently there are not that many details on scene graph construction, as the authors treat these as given. Are there specific properties of the scene graph that are important for enabling LLM-based planning? Does the amount of semantic information (e.g. object descriptions) matter for SayPlan performance?
6) The reliance on pre-built scene graphs that don't change is a major limitation, reducing applicability to dynamic or uncertain environments. The open-loop planning approach without real-time replanning based on perceptions is another drawback. The authors acknowledge both of these points, and understandably addressing both in the same paper is perhaps out of scope.

**Quality Of The Limitations Section:**

Limitations are addressed clearly

**Questions For Rebuttal:**

See feedback section

**Robotics Focus:**

Sufficient demonstration on hardware

**Summary Of Paper:**

This paper addresses the challenge of scaling task planning for robots to large, multi-room environments using large language models (LLMs). The SayPlan approach incorporates 4 key ideas: (1) Representing environments as 3D scene graphs that capture semantic and spatial relationships between objects and locations (2) Allowing the LLM to explore and focus on task-relevant subgraphs to avoid exceeding token limits. (3) Using a classical path planner to complete navigation actions, reduces the LLM's planning horizon. (4) An iterative replanning process with a scene graph simulator to ensure plan feasibility, where feasibility feedback is given to the LLM to iteratively improve its plans. Experiments in large multi-floor environments demonstrate the approach scales effectively and produces more executable plans than vanilla LLM task planning.

**Summary Of Recommendation:**

The paper is relevant and well-executed. Additional clarifying experiments on environment scale and iterative planning iterations would make the paper stronger.

---

> ### Author Response · Authors · 2023-08-14
> **Author Response**
>
> We thank the reviewer for the time they have taken to provide detailed and constructive feedback towards the betterment of this manuscript. We address each of the raised concerns and provide our detailed responses below across 2 parts. We also have attached the revised manuscript and have denoted all the new text and figures added in red.

---

> > ### Author Response · Authors · 2023-08-14
> > **Part 2**
> >
> > **There are no comparisons to task planners that do not use LLMs. Since the scene graphs are given, are there really no other task-planning methods that can serve as a reasonable baseline?**
> >
> > We agree with the reviewers that there are alternative classical approaches for task planning that could possibly leverage our 3DSG for task planning, however, these systems lack the generality and common sense capabilities of LLMs for task planning across a diverse range of tasks in large-scale environments, which was the primary motivation behind the instruction set used for evaluation. Adapting our evaluation towards these approaches would have required us to adapt our 3DSG to the required format that they require e.g. PDDL and semantically parse each instruction in order to generate a task-specific description that the traditional planner could utilise, which was beyond the scope of what we wanted to show with this paper. We have included a new statement in the related work section to motivate why we primarily focus on LLM-based planning techniques.
> >
> >
> > **Are there specific properties of the scene graph that are important for enabling LLM-based planning? Does the amount of semantic information (e.g. object descriptions) matter for SayPlan performance?**
> >
> > The semantic information attached to a node plays an important role in enabling the LLM to reason over the appropriate parts of the 3DSG with respect to the semantics of the query instruction. The types of semantics to include are heavily influenced by the type of task or level of detail required to perform a given task instruction. As a minimum requirement for the set of instructions evaluated in our work, we found it important to have meaningful node names that the LLM can relate to during Semantic Search e.g. kitchen, agriculture_lab, as well as appropriate attributes attached to these nodes that are not typically captured by the node name e.g. large, red, rotten. We provide specific examples in our Semantic Search evaluation where no meaningful semantics are provided e.g. Find object K31X  where we see the LLM resort to performing more of a breadth first-like search, blindly expanding all the nodes in the top level of the 3DSG hierarchy until it finds what it is looking for. We have now included an additional statement to highlight this importance in the Results section (Section 5.1).

---

> > ### Author Response · Authors · 2023-08-14
> > **Part 1**
> >
> > **More analysis could be provided on how the plan quality changes with increasing environment size.**
> >
> > We acknowledge the importance of a scalability analysis to improve the strength of the paper. Following the reviewer's suggestion, we have now included new experimental results in the revised manuscript under Appendix G which demonstrate the performance of SayPlan as we scale to larger environments. More specifically as SayPlan’s initial input is a contracted 3DSG, we performed an analysis on how the number of nodes present in this contracted 3DSG impacted both the Semantic Search and Iterative Planning stages of the pipeline. Across all the experiments, we found that the presence of additional and potentially “noisy nodes did not impact the ability of the underlying LLM to attend to the relevant components of the scene graph. All failures within this study resulted from the token limit overflow (~300 nodes in the highest level of the 3DSG), which in the case of the GPT-4 model we have available was 8192 tokens. With larger capacity LLMs, and appropriate hierarchical structures within the 3DSG, these results support our belief that SayPlan will continue to enable task planning across even larger-scale environments.
> >
> > **It might be good to see at which scale SayPlan starts to outperform the baselines.**
> >
> > We note here that SayPlan is to the best of our knowledge one of the first bodies of work that enables LLMs to plan over large environments spanning up to 3 floors, 36 rooms and 140 assets and objects. To this end, all our baselines had to be adapted to leverage our scene graph-based approach, in order to conduct a fair evaluation across the same sets of environments and instruction sets. Each baseline however lacked a key property of the SayPlan architecture, including the integration of a classical path planner and the ability to iteratively replan. Apart from the scene graph representation and manipulation via semantic search, these two components allowed our approach to scale to long-horizon task plans. With respect to the reviewer's comment at what scale does SayPlan start to outperform the baselines, we refer to the results in Table 4 and Appendix F which demonstrate that this particularly occurs when the horizon of the task plans increases, and the ability to facilitate planning with both a classical planner and LLM self-reflection become important factors for outperforming the two baselines we compared to.
> >
> >
> > **Ablations on plan performance vs. the number of iterations**
> >
> > We thank the reviewer for this suggestion and we acknowledge the importance of demonstrating the impact of iterative replanning on the planning performance of SayPlan. To this end, we have included a new set of results in Appendix F  that provides a detailed account of each causal planning query, including the number of iterative replanning steps it took for the agent to arrive at the final correctness and executability metric recorded in Table 4.

---

### Official Review · Reviewer_ZPG6 · 2023-07-19

**Confidence:** 5
**Originality:** Very Good
**Technical Quality:** Excellent
**Clarity Of Presentation:** Excellent
**Impact:** 4

**Recommendation:**

Strong Accept: I recommend accepting the paper and will argue for my recommendation even if other reviewers hold a different opinion.

**Review:**

The paper is very well written and the presented method is both well thought out and well executed. It is a clever idea that directly address a central limitation of LLM-based planning and is likely to spawn a new direction. The prose is clear. The paper on the whole is well organized and easy to follow. Results feel complete for this initial effort in this direction and raise interesting questions and ideas for future work. I have little in the way of suggestions to improve the paper.

The only "core issue" that I have with the paper is one that I do not entirely know how to solve: while the main body of the paper is self-contained and does a good job of describing the approach and presenting results statistics, individual instances of fully-executed plans as well as the back-and-forth dialogues with the LLM only appear in the appendix. While clearly this is too much information to include in the body of the paper, adding some elements of it would help build intuition useful for understanding the work.

For example, the discussion of Semantic Graph Search (Line 171) would benefit greatly from a snippet showing a single input/output from Appendix I. Even without much context, showing the form that "memory" and the 3d scene graph take for a *very* simple example would help readability. If only a very simple hand-crafted example were provided and included in a small figure on the right side of the page, hopefully it would take up only a small bit of space yet provide incredibly useful detail that Figure 1 lacks. This is similarly true of the Iterative Re-planning

Ultimately, the paper stands well enough on its own, but the impact of the paper will be enhanced if such details are readily available in the main body.

Other comments and suggestions:
- It is mentioned that there is a limit of 5 iterative replanning steps. Is there a similar limit placed on semantic search? Relatedly, how would SayPlan handle being given a task that could not be accomplished in the given environment?
- The bottom margin of page 2 is off somehow, as the text is touching the page number.
- Capitalization in the references is off for some entries: e.g., "Ai Magizine" in [26], "factored mdps in [27]". In BibTex, an extra set of {curly braces} is needed to preserve the capitalization of specific words. Please go through the references and address these changes.
- Many of the left-side quotation marks are facing the wrong way. In LaTeX, two back-ticks (``) should be used to indicate left-quotes.
- Curiosity Question: Is SayPlan currently able to handle object state, for example that a dishwasher is currently full and clean? Do nodes in the graph have state information other than their connectivity in the graph?


**Quality Of The Limitations Section:**

Limitations are addressed clearly

**Questions For Rebuttal:**

[Reproduced questions from the longer review above.]

- It is mentioned that there is a limit of 5 iterative replanning steps. Is there a similar limit placed on semantic search? Relatedly, how would SayPlan handle being given a task that could not be accomplished in the given environment?
- Curiosity Question (not essential): Is SayPlan currently able to handle object state, for example that a dishwasher is currently full and clean? Do nodes in the graph have state information other than their connectivity in the graph?


**Robotics Focus:**

Highly relevant to robotics but no hardware experiments

**Summary Of Paper:**

This paper presents SayPlan: a method for high-level task planning in large-scale fully known building-like environments in which LLM-based action proposal is supported by and verified against a 3D scene graph representation of the state of the world. The central approach consists of two parts, each focused on expanding/contracting nodes in a compactified representation of the map until the LLM context includes all objects needed for planning and then using this information to plan. The approach outperforms competitive baselines and proves capable of accomplishing complex long-horizon tasks given only the LLM and the 3D scene graph.

**Summary Of Recommendation:**

[Restated from above.] The paper is very well written and the presented method is both well thought out and well executed. It is a clever idea that directly address a central limitation of LLM-based planning and is likely to spawn a new direction. The prose is clear. The paper on the whole is well organized and easy to follow. Results feel complete for this initial effort in this direction and raise interesting questions and ideas for future work. I have little in the way of suggestions to improve the paper.

---

### Official Review · Reviewer_UUK5 · 2023-07-24

**Confidence:** 4
**Originality:** Very Good
**Technical Quality:** Good
**Clarity Of Presentation:** Fair
**Impact:** 4

**Recommendation:**

Strong Accept: I recommend accepting the paper and will argue for my recommendation even if other reviewers hold a different opinion.

**Review:**

Overall, I find the work of high quality and the capabilities demonstrated are impressive.

Strenghts:

1. The paper demonstrates planning over large environments

2. Their approach shows the ability to plan over relatively abstract and complex concepts

3. I particularly like the way they have combined classical planning with verification of plan feasibility over the scene graph

Doubts / Areas of improvement:

1. I found the manuscript a bit difficult to follow and lacked detail and clarity where I would have wanted. This were clarified by digging through the appendices but I would suggest to move some things into the main manuscript I have a few suggestions for ways that it could be better:
 - Be more precise about what is the "scene graph simulator" and what the operations "collapse", "expand", and "contract". These terms are used very often and a key aspect of the approach but in my opinion never precisely defined. For example, the paragraph on "Semantic Graph Search" (L171 to L188) is insufficient to really understand the method.
 - I was confused during the reading of the manuscript about the "prompt" generation part, and what are the inputs and expected outputs from the LLM until I consulted the Appendices. This could be made more precise.
- I found Fig. 1 very difficult to parse. It is more easily understandable on the website where it is animated but in the manuscript it overwhelmingly complex.
- I think it could really help to have a running example to explain the various components.

 2. I would like the authors to justify their choice of baselines. In particular, they have not compared against any non "scene graph" based approaches. For example "Palm-SayCan" ("Do As I Can, Not As I Say: Grounding Language in Robotic Affordances") or other reinforcement learning style baselines. It is also a bit problematic that the specific queries that are used for evaluation are hand-engineered. Perhaps it is a byproduct of the fact that this field is a bit new, but it seems like there is a need for some standardization of the evaluation baselines here.

 3. The paper is clear and explicit about the assumption of having access to the ground truth 3D scene graph, but I could have used more clarity about what is assumed for robot skill execution. For example, for navigation, is there a full metric SLAM map that is used? How are the policies for manipulation trained? Etc.


Minor comments:

 - "feedback" is not initialized in Algorithm 1
 - The caption for Table 2 confuses me because it refers to an "average number of steps" but the numbers in the table are all percentages which I don't know how to interpret.

Some typos:

L32: as well AS comprehend
L151: Using 3D scene graph representation (no s)



**Quality Of The Limitations Section:**

Additional details required

**Questions For Rebuttal:**

See points 2 and 3 in the "Doubts/Areas of Improvement" above.

**Robotics Focus:**

Sufficient demonstration on hardware

**Summary Of Paper:**

The paper proposes a method based on large language models and 3D scene graphs for a robot to solve long horizon tasks. The approach has two main components: First, the natural language instruction is used to guide a semantic search through the scene graph. Second, an iterative planning method finds feasible plans with a classical planner and then verifies their correctness in the scene graph. Their results, demonstrated on a real mobile manipulator in a fairly large scale envorment, show examples of successful task executions.

**Summary Of Recommendation:**

Overall I find the concepts and demonstration in the work impressive (subject to getting satisfactory responses to the concerns I have raised). The paper is likely to have a significant impact in the field and therefore should be accepted.

---

### Decision · Program_Chairs · 2023-08-30

**Decision:**

Accept (Oral)

**Comment:**

The paper proposes a method that addresses the challenge of scaling task planning for robots to large, multi-room environments, using large language models (LLMs) in conjunction with 3D scene graphs. This is in part enabled by semantic search of task-relevant subgraphs to encode world model information to the LLM, and iterative re-planning with a classic planner in-the-loop to find feasible action plans.

The approach presents itself as an outstanding combination of both search (through structured representations) and learning (via LLMs) to address a challenging problem, with compelling real-world experiments on a mobile manipulator in large-scale environments.

Post-rebuttal, all reviewers unanimously agree on "Strong Accept." Excellent work!